# Global and regional sea-surface temperature changes over the Marine Isotopic Stage 9e and Termination IV

Nathan Stevenard<sup>1</sup>, Émilie Capron<sup>1</sup>, Étienne Legrain<sup>1,2,3</sup> & Claire Coutelle<sup>1</sup>

Correspondence to: Nathan Stevenard (nathan.stevenard@univ-grenoble-alpes.fr)

<sup>&</sup>lt;sup>1</sup>Université Grenoble Alpes, CNRS, IRD, Grenoble INP, IGE, Grenoble, France.

<sup>&</sup>lt;sup>2</sup>Laboratoire de Glaciologie, Université libre de Bruxelles, Brussels, Belgium.

<sup>&</sup>lt;sup>3</sup> Department of Water and Climate, Vrije Universiteit Brussel, Brussels, Belgium.

**Abstract.** The Marine Isotope Stage (MIS) 9e, occurring approximately from 335 to 320 ka, represents an important period for studying the dynamics of Earth's climate. Interest in studying this interglacial period stems from the fact that it is associated with the highest atmospheric CO<sub>2</sub> concentrations over the last 800 ka (excluding anthropogenic CO<sub>2</sub> emissions). Numerous reconstructions of sea surface temperatures (SST) cover this time interval, yet synthesizing them into consistent regional- and global-scale climate signals is challenging because they are scattered across the globe and based on heterogeneous chronological frameworks. In this study, we present the first spatio-temporal SST synthesis over the interval 350 to 300 ka, covering this interglacial period and its preceding deglaciation (Termination IV, ~350 to ~335 ka). We include 98 high-resolution SST reconstructions and we establish a common temporal framework between the selected marine records, based on the latest reference ice core chronology (AICC2023). We also homogenize the proxy-calibration strategy by applying a single method for each proxy. Chronological and calibration uncertainties are quantified using Bayesian and Monte Carlo procedures. Finally, through a Monte Carlo approach, we generate global- and regional-scale SST stacks relative to Pre-Industrial Era over Termination IV and MIS 9.

We highlight significant differences in terms of temporal variability, amplitude, and timing of changes in the SST records across the globe over the studied time interval. While the patterns of SST changes are homogeneous at basin-scale, heterogeneous interglacial SST peaks are observed across ocean basins. The interglacial surface temperature peaks in extratropic basins are similar to or warmer than the pre-industrial period (PI), while intra-tropic areas appear to be colder relative to PI during interglacial optimum. In addition, the timing in interglacial surface temperature peaks differ across the different regions. These regional temperature variations suggest that atmospheric and oceanic dynamics played a greater role than global radiative forcing in shaping the MIS 9e climate. The heterogeneous timing of changes across the different regions contribute to a smoothed global-scale response in terms of both timing and amplitude. Consequently, we find that at a global scale MIS 9e SST was as warm as the PI ( $\sim$  -0.1°C  $\pm$  0.2 °C). Converted into surface air temperatures ( $\sim$  -0.3°C  $\pm$  0.3 °C), this estimate agrees within the uncertainty range with previous studies based on a smaller number of records with lower temporal resolution. We also compare our results on MIS 9e and Termination IV with published SST syntheses covering more recent interglacial periods (MIS 5e and Holocene) and deglacial periods (Termination I and II). We find that the global deglacial surface air warming during Termination IV is similar in amplitude ( $\sim$ 5.7 °C) to that observed during Terminations I and II. Finally, a comparison of deglacial warming rates for these three terminations to the warming trend of the last 60 years emphasizes that the rapidity of modern climate change is unprecedented within the context of these past deglaciations.

#### 1 Introduction

40

45

Past interglacials represent the warm periods of the Quaternary, sometimes exhibiting conditions as warm or warmer conditions than during the pre-industrial (PI. Past Interglacials Working Group of PAGES, 2016). Therefore, studying these intervals is helpful to better understand the interactions between the different components of the climate system in a range of temperature comparable to projected future changes (Capron et al., 2019). While the global or regional temperature variability is well constrained for the most recent interglacials such as the Holocene (11 - 0 ka: 1 ka = 1000 years; e.g. Shakun et al., 2012; Osman et al., 2021) or the Last InterGlacial (LIG, 129 - 116 ka; e.g. Capron et al., 2014, 2017: Hoffman et al., 2017), multi-millennial-scale global temperature changes for older interglacial periods are not available. Particularly, the Marine Isotopic Stage (MIS) 9e (335 to 320 ka) stands out as one of the warmest interglacial of the last 800 ka (Past Interglacials Working Group of PAGES, 2016). It is characterized by the highest atmospheric CO<sub>2</sub> (300.4 ppm) and CH<sub>4</sub> (818 ppb) concentrations levels as recorded in the EPICA Dome C over the last 800 ka (Bereiter et al., 2015; Loulergue et al., 2008; Nehrbass-Ahles et al., 2020). Relative Sea Level (RSL) estimates for MIS 9e remain poorly constrained, with values ranging from  $-10 \pm 13$  m, estimated with a Red Sea record based on a  $\delta^{18}$ O record (Grant et al., 2014), to  $-1 \pm 23$  m from global  $\delta^{18}$ O benthic foraminifera records (Spratt and Lisiecki, 2016), Other discrete RSL data based on coral estimate a range from -20 to -3 m (Medina-Elizalde, 2013), Additionally, Termination IV (T-IV), the deglaciation preceding MIS 9e, is marked by an exceptionally rapid sea-level rise (~4.9 m per 100 years compared to a rise of less than 3 m per 100 years for T-I and T-II), the highest of the last 500 ka (Grant et al., 2014). These factors make MIS 9e and T-IV a relevant time interval for studying climate responses to naturally high GreenHouse Gases (GHG) concentrations and rapid sea-level rise.

Despite this relevance, our understanding of global or regional temperatures during MIS 9e and T-IV still relies mostly on reconstructions derived from long-term temperature stacks (e.g. Shakun et al., 2015; Snyder, 2016; Friedrich et al., 2016; Clark et al., 2024). For instance, Shakun et al. (2015), compiling 49 SST records over the past 800 ka, estimate that the global surface ocean temperature during the MIS 9e peak was slightly warmer (~1.8°C) than the late Holocene, with a deep ocean temperature ~2°C warmer than Holocene values (the warmest interglacial peak over the last 800 ka). Similar estimate of the Mean Ocean Temperature (MOT) at MIS 9e was derived from the noble gas composition of the air trapped in the EPICA Dome C ice core (Haeberli et al., 2021). More recent estimates of global surface temperature indicate a MIS 9e peak close to PI conditions, with estimates of  $0.4 \pm 2.2$  (2 $\sigma$ ; Snyder, 2016),  $0.1 \pm 0.6$  ( $\sigma$ ; Clark et al., 2024) or ~ -0.5 ± ~2 °C ( $\sigma$ ; Friedrich et al., 2016) compared to the PI. However, all these global surface temperature reconstructions are based on a relatively small number of records (~20–35 records) and most of them have low temporal resolution. Also, they often rely on imprecise chronologies derived from alignments of the benthic  $\delta^{18}$ O record at a given site to the LR04 benthic  $\delta^{18}$ O stack (Lisiecki and Raymo, 2005) which was dated through orbital tuning and is associated with relatively large absolute age uncertainties ( $\pm$  4 ka). Also, focusing on the time interval covering MIS 9 and Termination IV only, more high-resolution surface temperature records are available than those included in the compilations covering longer time-scales. However,

those paleoclimatic records have been dated using various climatic alignment strategies. Those limitations related to the construction of the paleorecord chronologies (discussed in length in Govin et al., 2015) prevents the investigation of the spatio-temporal structure of surface temperature changes at multi-millennial-scale over MIS 9e and the preceeding Termination IV.

75

100

Beyond chronological uncertainties, the variety of SST proxies and the improvement in proxy-calibrations over decades challenge a direct comparison between published records. The existing SST reconstructions covering the MIS 9e have been produced using a wide range of proxies (see Table S1), each with distinct characteristics and temperature sensitivities. The alkenone unsaturation index  $(U_{37}^{K\prime})$  is based on the relative unsaturation of long-chain  $(C_{37})$  alkenones, which decreases with increasing temperature (Prahl et al., 1988). It typically reflects annual SST, but may capture seasonal temperatures at high latitudes (~45°N) or in semi-enclosed basins such as the Mediterranean (Tierney and Tingley, 2018). The Modern Analogue Technique (MAT) estimates SST by comparing fossil planktic assemblages (e.g. foraminifera, radiolarians) with modern datasets using a transfer function, identifying the closest analogues based on species compositions (e.g. Ruddiman et al., 1989). MAT generally reflects seasonal SST. The magnesium-to-calcium (Mg/Ca) ratio of planktic foraminifera reflects a nonlinear increase in magnesium incorporation into calcite shells with rising temperature (Oomori et al., 1987). While often associated with mean annual SST, it may reflect seasonal conditions depending on the foraminifera species and the site's latitude (Tierney et al., 2019). The  $\delta^{18}$ O from planktic foraminifera ( $\delta^{18}$ O<sub>p</sub>) is affected by both temperature and the isotopic composition of seawater ( $\delta^{18}O_{sw}$ ) at the time of calcification. Although calibration is complex due to varying relationships derived from culture and plankton tow studies, recent Bayesian approaches (Malevich et al., 2019) allow for SST estimates that incorporate uncertainties in  $\delta^{18}O_{sw}$  and can reflect either annual or seasonal SST, depending on species and calcification latitude (Malevich et al., 2019). The  $TEX_{86}$  (TetraEther indeX of 86 carbons) proxy is based on the relative distribution of archaeal glycerol dibiphytanyl glycerol tetraether (CDGT) lipids produced by marine archaea (Schouten et al 2002). It can estimate SST through a temperature-dependent change in ring structures, but often reflects subsurface conditions (e.g. Schouten et al 2002; Lopes Dos Santos et al., 2010; Tierney & Tingley, 2014; Rouver-Denimal et al., 2023), limiting its comparison with surface-based SST reconstructions. Recent improvements in proxycalibrations (e.g. Tierney and Tingley, 2014, 2018; Malevich et al., 2019; Tierney et al., 2019) have shown substantial differences compared to earlier methods. For example, Mg/Ca-derived SST can vary by several degrees depending on the cleaning method or pH correction used (Gray and Evans, 2019). These discrepancies highlight the need to recompute existing SST records with harmonized and updated calibration tools to enable consistent comparisons.

To summarize, the lack of temporal precision combined with the low number and resolution of aligned records from the existing SST syntheses of MIS 9e currently prevents a detailed description of multi-millennial-scale spatio-temporal climate changes and limits the exploration of the mechanisms involved, such as previously done for younger intervals (e.g. Shakun et al., 2012; Stone et al., 2016; Hoffman et al., 2017; Osman et al., 2021; Gao et al., 2024). In this study, we present a new high-resolution SST synthesis covering MIS 10, T-IV and MIS 9 (300–350 ka) based on 98 high-resolution (

**Figure 1: Location and latitudinal distributions of annual SST records.** (A) Cumulative number of records per 5° latitudinal bin. (B) World map with the different types of proxies and their location. Proxies are  $U_{37}^{K\prime}$  (yellow), δ<sup>18</sup>O<sub>p</sub> (purple), Mg/Ca ratio (pink) and MAT (dark red) for seasonal (diamonds) and annual (circles) SSTs.

#### 2.2 SST calibrations







Based on recent advances in SST calibrations, we recalibrate the original data using a Bayesian approach which better represents the uncertainties associated with each proxy's specificities. This approach was previously applied in SST syntheses for the recent period of time (24 to 0 ka, Osman et al., 2021; 21 to 18 ka, Tierney et al., 2020).

To calibrate the  $U_{37}^{K\prime}$  data (23 records), we used the BAYSPLINE Matlab package (Tierney and Tingley, 2018). Slope attenuation is an important feature in high-temperature areas, where the  $U_{37}^{K\prime}$  data saturate near a value of 1. Therefore, the BAYSPLINE Bayesian tool takes these features into account to produce more realistic SST reconstructions. The algorithm first calculates the SST on the basis of the Prahl et al. (1988) calibration to define the prior mean. Following Osman et al. (2021), we applied a prior standard deviation of 5°C. The BAYSPLINE package then produced a  $N \times 1000~U_{37}^{K\prime}$  matrix of SST possibilities for each age N.

The Mg/Ca calibrations (17 records) were done with the BAYMAG Matlab package (Tierney et al., 2019). Since Mg/Ca is a complex paleo-thermometer due to its sensitivity to multiple environmental factors such as the calcite saturation state ( $\Omega$ ), salinity, or seawater pH, the original calibration between Mg/Ca value and temperature (e.g., Anand et al., 2003) is outdated. A key advantage of using the BAYMAG calibration is that all environmental sensitivities (if known) are included in the Bayesian model, avoiding the need for pre-correction of the Mg/Ca data. In this study, the prior mean is automatically defined by the model as the mean SST from an initial calibration with Anand's model (Anand et al., 2003). Following Osman et al. (2021), we applied a prior standard deviation of 6°C. The pH and salinity estimates were based on a modified function from Gray & Evans (2019), using the benthic  $\delta^{18}$ O stack LR04 (Lisiecki and Raymo, 2005) as sea-level change reference. The modern calcite saturation state ( $\Omega$ ), pH, salinity, and temperature values were defined using two functions inherent to the BAYMAG package. The sample cleaning technique and the foraminifera species used for Mg/Ca measurements were defined based on published information. Once all the prior information was defined, we ran the BAYMAG model. This model produces an  $N \times 2000$  matrix of Mg/Ca SST possibilities for each age N, then randomly subsampled to produce a smaller matrix with dimensions of about  $N \times 1000$  to be in line with the other proxies.

The  $\delta^{18}O_p$  calibrations (37 records) were carried out with the BAYFOX Matlab package (Malevich et al., 2019). This package includes hierarchical models for annual, seasonal, and/or species-specific calibrations. Since we did not have sea water  $\delta^{18}O$  (hereafter called  $\delta^{18}O_{sw}$ ) estimates during the MIS 9 for all sites, we used the modern  $\delta^{18}O_{sw}$  from Breitkreuz et al. (2018) to run these calibrations. A simple ice volume correction following Malevich et al. (2019) was applied before calibration, using the benthic  $\delta^{18}O$  stack LR04 (Lisiecki and Raymo, 2005) as a reference for global ice-volume changes. Finally, the prior mean was estimated either using other published SST records from the same core (published mean) if available, or using the Pre-Industrial (PI; 1870-1889) SST from HadISST (Rayner et al., 2003) as the prior mean. A fixed

prior standard deviation of 10°C was applied as done in Osman et al. (2021). The ensemble of probable SSTs derived from  $\delta^{18}O_p$  is a  $N \times 1000$  matrix.

To our knowledge, no Bayesian approach has been developed for MAT calibrations. Most of the time, the distribution of foraminifera species is not published and only the derived SST record is available. Therefore, to propagate the uncertainties of MAT data (21 records), we conducted a Monte-Carlo analysis, randomly creating 1000 data points for each data point, following a normal distribution  $N(\mu,\sigma)$ . If this error  $(\sigma)$  was not published in the original publications, we estimated it as the standard deviation of the SST record over the period 300-350 ka. After 1000 realizations for each SST data point, the ensemble of MAT data also forms a  $N \times 1000$  matrix.

For each proxy, the SST type (seasonal or annual) is defined based on the recommendation of the original studies or inferred from the calibration specifications. As a result, this synthesis comprises 74 records of annual SSTs and 20 records of seasonal SSTs.

#### 2.3 Anomaly from the pre-industrial period

In existing SST syntheses, the SST records are commonly transferred into "anomaly" compared to a reference (e.g. Capron et al., 2014; Snyder, 2016 Hoffman et al., 2017; Tierney et al., 2020; Osman et al., 2021; Clark et al., 2024). In this study, we define the SST anomaly from the PI period. Since core-tops are rarely well preserved in our SST records and to harmonize the creation of anomalies, we defined the PI-SST (1870 to 1899 CE as used in Capron et al., 2017) by forward modelling (with the Bayesian calibrations previously described) the SST from the HadISST database (Rayner et al., 2003). This step gives  $1 \times 1000$  probable proxy-values derived from the SST database. We then converted our modelled proxy-value (median value) in a  $1 \times 1000$  ensemble of PI SST. For MAT reconstructions, an ensemble of  $1 \times 1000$  probable PI SST values is generated from a normal distribution using the mean SST and standard deviation derived from the HadISST database (Rayner et al., 2003). This approach allows a better estimate of the proxy-SST value, which can differ from the HadISST database (Rayner et al., 2003). The same process was applied to seasonal proxies, taking only the corresponding monthly mean SST from the HadISST database. However, we are aware of the limitation of this PI reference and discrepancies between model ensembles and the HadISST database can exist, especially in the Southern Ocean (Gao et al., 2024). The anomaly from the PI is defined, for each record, by sorting the N<sup>th</sup> ensemble of SSTs (1000 values) from least to greatest and subtracting the PI (also sorted from least to greatest).

#### 2.4 Chronologies






In this study, we harmonize the chronologies across all marine sediment records. Due to the inability to constrain the age model with radiocarbon dates, as for the syntheses covering the most recent periods (Tierney et al., 2020; Osman et al., 2021; Gray et al., 2023), the revision of original chronologies for periods older than ~50 ka must rely on alternative strategies (Govin et al., 2012; Capron et al., 2014; Hoffman et al., 2017). For records located in high latitude areas (i.e., >

40° North or South), Govin et al. (2012) demonstrated that SST can be directly aligned with surface air temperature (SAT) records over Antarctica or Greenland for the last interglacial (LIG). For the same period, Hoffman et al. (2017) produced a new SST synthesis by aligning the benthic foraminifera  $\delta^{18}$ O records to "basin references", which themselves had revised chronologies based on the alignment of SST changes to ice-core SAT variations.








In this study, a similar approach was used. As for the Govin et al. (2012) study, we first (step 1: **Fig. 2**) align the SST of four "basin reference records" from high latitudes to ice-core SAT with "AnalySeries" software (Paillard et al., 1996). All "basin references" (**Table S1**) contain published high-resolution SST reconstructions and benthic  $\delta^{18}$ O records. Since no direct reconstructed SAT are available for Greenland from deep ice cores beyond ~130 ka, we propose to use the Greenland synthetic curve of temperature (GL<sub>T</sub> syn, Barker et al., 2011). This curve is produced based on the millennial and long-term signal of Antarctica temperature records, with a 2 ka shift in order to account for the effects of the "bipolar seesaw" observed between Greenland and Antarctic ice core records over the last glacial period (see Barker et al., 2011). In Capron et al. (2014), the CH₄ record was proposed as an indirect tracer for Greenland climate. Comparisons of the GL<sub>T</sub> syn to the CH<sub>4</sub> record (Loulergue et al., 2008) indicates similarities at the first order, but also some discrepancies at millennial time-scales (Fig. S1). These millennial-scale differences are at least partly associated with the fact that, on one side the terrestrial biosphere response to hydroclimatic changes in lower latitudes may affect the CH<sub>4</sub> concentrations, and on the other side that the GL<sub>T</sub> syn is not strictly an indicator of past Greenland temperature as illustrated with the differences observed when compared to the water isotope record from Greenland NEEM ice core over the LIG (Govin et al., 2015). With those different biases in mind, we propose to use the GL<sub>T</sub> syn as reference of northern high latitude temperatures, as this strategy has been already used to align marine records to ice core chronologies on older timescales (e.g. Barker et al., 2015; Hodell et al., 2015, 2023).

The first step thus consists in the alignment of the millennial-scale variations of SST of high latitude sites (the basin references; **Table S1**) to those of the Greenland synthetic temperature curve (Barker et al., 2011) and reconstructed SAT from the  $\delta D$  of EPICA Dome C (EDC, Jouzel et al., 2007), both on the most recent ice-core chronology AICC2023 (Bouchet et al., 2023). Four "basin references" were defined, based on their high-resolution and the ability to align those records to reference records (i.e. ice-core records are synthetic curve). For the U1429 site (Northwest Pacific reference; **Table S1**), we aligned the variation of the  $\delta^{18}O_{\text{notched}}$  (i.e.  $\delta^{18}O$  of planktic foraminifera after removing the eccentricity- and obliquity-band variance; Clemens et al., 2018) of planktic foraminifera *G. ruber*, a proxy of the East Asian monsoon, to the  $\delta^{18}O_{\text{calcite}}$  stack (Cheng et al., 2016) from Sanbao cave. This  $\delta^{18}O_{\text{calcite}}$  record was previously scaled to the AICC2023 chronology by simple linear interpolation using the tie-points defined by Bouchet et al. (2023). Once "tie-points" were defined (**Fig. S2**), we visually estimated the depth and age uncertainties (from a few hundreds to thousands of years) to encompass the millennial-scale events as recorded in SAT references. Tie-points were defined using peaks and troughs, as this approach provided the best visual alignment with the reference records. We acknowledge that such a method may risk overfitting, particularly in lower-resolution records, where the identification of peaks and troughs is less robust. However, because we rely on age ensembles rather than single age models, the impact of this choice on the final results is minimized. The depth uncertainties

are, most of the time, equivalent to the depth resolution of the record. The age uncertainties are most of the time estimated as half of the duration of the event (peak or trough). At this point, only basin references are aligned to the AICC2023 ice-core chronology (**Fig. S2**).

The second step is then to align the other sites, using both basin references and ice-core records. Three scenarios occur: (1) the site has a published benthic  $\delta^{18}$ O record with sufficient resolution and is aligned to a basin reference (29 sites); (2) the site has no benthic  $\delta^{18}$ O record, is located in the "high latitude areas" and is aligned to ice-core SAT reconstructions (15 sites); (3) the site has no benthic  $\delta^{18}$ O record, is not located in high latitudes and is finally aligned to SST from basin references or the nearest site already aligned with the first or second method (15 sites). For each method, the age and depth uncertainties are visually defined as described above.

The third and last step consists in running a first Bayesian age-depth model with the "*Undatable*" Matlab software (Lougheed and Obrochta, 2019) to estimate the alignment uncertainties. This software takes into account the sedimentation rate uncertainties by adding a new parameter "*xfactor*" and randomly bootstraps age-depth tie-points with the "*bootpc*" parameter. As this first bayesian age model is only to estimate the alignment uncertainty on the basis of both depth and age visual errors, we ran these Bayesian age-depth models by simulating 10<sup>5</sup> chronologies, with an *xfactor* of 0.1 and a no bootstrap. Following this step, we estimate the absolute age uncertainty, for each site, by calculating the quadratic sum of the alignment uncertainty, and the references uncertainties which were used. As an example, the absolute uncertainties of a site aligned to the basin reference "IODP U1429" include the site alignment uncertainties of each tie-points, the U1429 to Sanbao alignment uncertainties, the Sanbao to AICC2023 uncertainties, and the absolute uncertainties of the AICC2023 ice-core chronology. Then, we run the final Bayesian age model using "*Undatable*" (Lougheed and Obrochta, 2019), with 10<sup>5</sup> simulations, a "*xfactor*" of 0.1 and 15% of tie-points bootstrapped in each iteration.

**Figure 2: Description and examples of the chronology construction.** In step 1 and 2, the blue curve is the record used to align to the reference (black curve). Dashed vertical lines in "Step 1" and "Step 2" represent the tie points used to align the record to the reference. For the North Atlantic example, the  $\delta^{18}O_p$  from core MD01-2443-2444 (Martrat et al., 2007) is aligned to the GL<sub>T</sub>\_syn (Barker et al., 2011), and the  $\delta^{18}O_p$  record from ODP-980 (McManus et al., 1999) is align to the MD01-2443-2444 one with the revised age scale.

As a result, the prior mean quadratic sum of all the sites range from  $\pm$  1.2 to  $\pm$  4.8 ka of uncertainties, with a mean error for all sites of  $\pm$  2.7 ka. All Bayesian age-depth models are reported in **Fig. S3**. For each depth-SST pair (*N*), we produce an ensemble of 1000 probable ages to have a  $N \times 1000$  matrix.

### 2.5 Global and regional stack constructions







These two Bayesian processes produce ensembles ( $N \times 1000$  possibilities) of both ages and SSTs for each depth-proxy pair (N) for every record. To estimate a Global Sea Surface Temperature anomaly (called  $\Delta$ GSST hereafter) during MIS 5e, Hoffman et al. (2017) used a Monte-Carlo procedure. They gridded their records (SST anomalies from the PI) into a 5×5° grid and calculated the area-weighted mean and stacked 1000 Monte-Carlo iterations to construct the final global stack interpolated at regular time steps (500 or 1000 years). However, their procedure has two major drawbacks: (1) the PI from the HadISST database (Rayner et al., 2003) does not reflect the SST from a proxy and have not any uncertainties as it is a single fixed value; (2) the non-uniform geographic distribution of their records induces a bias in the global mean temperature if a large number of grids are concentrated into a single latitudinal band.

Hence, we propose here to follow an alternative strategy for computing the ΔGSST for MIS 9. Our approach is largely inspired by Osman et al. (2021) but with a few differences. The global and regional stacking processes are based on the annual SST anomaly records with their revised chronologies (see section 2.3). In every step of this process, we use the ensemble of data (i.e. the 1,000 probable ages and SST for each age-SST pair) to better preserve sources of uncertainties. The first step of the Monte-Carlo process consists of randomly resampling the Nth age-SST ensembles to generate new age-SST pairs in each iteration. The second step, applied to each 500-year time bin, we first compute site means (averaging all data from ensembles available for this time-bin within a site, regardless of the proxy). These are then aggregated into gridcell means, using randomly defined grid sizes (2 - 5°), and subsequently into weighted latitudinal means, using bands of randomly varying width (2.5 - 10°). These processes reduce the impact of the spatial distribution of the sites, and the random factor provides an opportunity to account uncertainties in the spatial "choice" of the grid and latitudinal bands. In the third step, we define the  $\Delta$ GSST as the mean of all latitudinal band averages and scaled it as Global Mean Surface Temperature (i.e. "air" temperature, ΔGMST) by applying a random factor between 1.5 and 2.3 (Snyder, 2016; Tierney et al., 2020; Osman et al., 2021). These values come from a comparison between  $\Delta GSST$  and  $\Delta GMST$  from glacial maximum climate states as simulated by climate models (Snyder, 2016). This Monte-Carlo process was repeated 10,000 times to propagate errors. At each iteration, regional stacks were also created. These stacks are based on a hemispheric scale (North, Tropics and South) or basin scale (North Atlantic, North Pacific, Equatorial Pacific, South Atlantic, Indian Ocean and South Pacific).

#### 3 Results





#### 3.1 Global and regional sensitivity tests

Based on the individual description of each records (Supplementary Information S1), we identified some discrepancies between SST reconstructions at a same site (i.e. with different proxies) or within the general tendency of a region. Most of these inconsistencies are linked to  $\delta^{18}O_p$ - or, to a lesser extent, Mg/Ca-based SST, suggesting a lack of environmental context prior to the calibration (see section 4.1.1 for further discussions). To evaluate the impact of such records on the stacking procedure, we conducted three sensitivity tests: in the test (1), we include all annual SST records (n=77); in the test (2), we exclude all  $\delta^{18}O_p$ -based SST reconstructions (n=41), which often display lower temperature values or distinct pattern of variability; in the test (3), we retain only records showing a consistent range and pattern of variability of records located in the same area, leading to the exclusion of 13 records (n=64) primarily based on  $\delta^{18}O_p$  and Mg/Ca proxies. The selection in this third test is based on a visual assessment.

Overall, the resulting stacks display very similar variability across tests, regardless of the method applied. At global and hemispheric scales, the amplitude and timing of major climatic features (e.g. Deglaciation, interglacial optimum, glacial inceptions) remain unchanged (**Fig. 3**). At basin scale, however, the South Atlantic and Indian stacks without  $\delta^{18}O_p$ -based SST records (second test, **Fig. 3H, J**) show differences in the shape of variability. The main differences between tests are related to the absolute temperature values and amplitude: stacks excluding  $\delta^{18}O_p$  records are systematically warmer (~1 to 2 °C), with larger hemispheric amplitudes, and the North Atlantic shows particularly cold MIS 10 conditions in this case. The test including all records (**Fig. 3A-E**) is broadly consistent with the others but results in lower SST values, indicating that inconsistent records tend to pull down the stacks and potentially bias the reconstructions. Excluding  $\delta^{18}O_p$ -based SST (second test, **Fig. 3F-J**) produces the largest shift in values, but also strongly reduces the number of records. The "selected records" test (**Fig. 3K-O**) appears to provide a reasonable compromise: it excludes outlier estimates without systematically discarding one proxy type, while retaining a sufficient number of records to build robust stacks. We therefore adopt this third stacking approach for the analyses and interpretations that follow.

Figure 3: Sensitivity tests about the stacking method. (A-E) stacks realized with all annual records; (F-J) stacks without δ<sup>18</sup>O<sub>p</sub>-based SST records; (K-O) stacks with selected records (13 were excluded). (A, F, K) ΔGSST (purple) and ΔGMST (dark gray) stacks; (B, G, L) hemispheric ΔSST stacks with the NH (blue), the tropical band (black) and SH (red) anomalies; (C, H, M) Atlantic ΔSST with the North (blue) and South (red) Atlantic anomalies; (D, I, N) Pacific ΔSST with the North (blue), the Equatorial band (black) and South (red) Pacific anomalies; (E, J, O) Indian Ocean (red) anomalies. Shaded envelopes represent each percentile around the median (solid line).

#### 3.2 Global and regional SST changes


The following description relies on the global and regional SST stacks (see **section 2.5** and **3.1**). The different sources of uncertainties related to the chronology, the SST calibration and the spatial distribution of the records are accounted for in the construction of the global and regional stacks. Hence, the dates indicated in the following sections

represent the most probable dates obtained by combining all these different source of uncertainties. Errors in Celsius degree are given as the  $\sigma$  error.

#### 3.2.1 Global scale







On a global scale, the  $\Delta$ GSST and  $\Delta$ GMST (**Fig. 4C**) are marked by well-defined glacial conditions at around 348 ka with temperatures that are respectively 3.2  $\pm$  0.2 °C and 6.1  $\pm$  0.9 °C cooler than during the PI. This cold period is followed by a ~14 ka-long continuous warming in two steps: there is a first phase associated with a slow warming rate, and then, a second phase with an increased warming rate starting at 341.5 ka and marking the onset of the deglaciation (i.e. when the global temperature starts to rise sharply). The amplitudes of the deglacial warming are about ~2.4 and ~4.5 °C, respectively for  $\Delta$ GSST and  $\Delta$ GMST (**Fig. 4C**). The climatic optimum starts at 333  $\pm$  0.25 ka and extends up to 328.5  $\pm$  0.25 ka. During this period, the maximum  $\Delta$ GSST and  $\Delta$ GMST are respectively -0.1  $\pm$  0.2 °C and -0.3  $\pm$  0.3 °C relative to the PI (**Fig. 4C**).

This climatic optimum is followed by a 9 ka-long continuous cooling characterized by a  $\Delta GSST$  and  $\Delta GMST$  decrease of respectively 1.3°C and 2.2°C to reach values of -1.3  $\pm$  0.2 °C and -2.5  $\pm$  0.5 °C relative to PI. After a 2,500-years-long period of stability, the  $\Delta GSST$  and  $\Delta GMST$  exhibit a millennial-scale warming of reduced amplitude (~0.2 and ~0.5°C for  $\Delta GSST$  and  $\Delta GMST$ , respectively). Global temperatures then show a brief period of stability (~2  $\pm$  0.25 ka), after which they continuously decrease until the end of the studied period (i.e. 300 ka). This final cooling, however, does not reach the MIS 10 glacial range of temperature values (**Fig. 4C**).

#### 3.2.2 Hemispheric scale

On a hemispheric scale (**Fig. 4D**), the MIS 10 glacial temperature  $\Delta$ SST values are colder in the extra-tropic Northern Hemisphere (NH; -3.9 ± 0.7 °C) compared to the extra-tropic Southern Hemisphere (SH; -3.4 ± 0.4 °C). Tropical (between 23° North and South) area does not exhibit a distinct glacial maximum, and the colder temperature anomaly is -3.1 ± 0.2 °C at 350 ka. The hemispheric  $\Delta$ SST stacks exhibit differences in shape, amplitude, and timing of the deglacial warming. In the SH (**Fig. 4D**), the deglaciation starts at 346 ± 0.25 ka, lasts approximately ~11 ka and has an amplitude of warming of ~3.2 °C. In the NH, the deglacial warming begins at 342 ± 0.25 ka, lasting for approximately ~9 ka, with an amplitude of ~4.5 °C (**Fig. 4D**). The Tropics, shows a deglacial warming at 338.5 ka and lasts ~8 ka, with a lower amplitude of warming of ~1.9 °C (**Fig. 4D**).

**Figure 4: Global and regional temperature stacks over MIS 9.** (A) Annual insolation anomalies from the 0-1 ka BP mean (from Laskar et al., 2004); (B)  $CO_2$  composite from Bereiter et al. (2015) and Nehrbass-Ahles et al. (2020); (C)  $\Delta GSST$  (purple) and  $\Delta GMST$  (dark gray) stacks; (D) hemispheric  $\Delta SST$  stacks with the NH (blue), the tropical band (black) and SH (red) anomalies; (E) Atlantic  $\Delta SST$  with the North (blue) and South (red) Atlantic anomalies; (F) Pacific  $\Delta SST$  with the North (blue), the Equatorial band (black) and South (red) Pacific anomalies; (G) Indian Ocean (red) anomalies. Shaded envelopes in C to G represent each percentile around the median (solid line).

As a result, the interglacial conditions are reached at 335, 333 and 331 ka in the SH, the NH and the Tropics, respectively. The maximum temperature anomaly during the interglacial peak is slightly higher in the NH and the SH to those observed for the PI period, with respectively  $0.6 \pm 0.5$  and  $0.1 \pm 0.3$  °C (**Fig. 4D**). However, in Tropics, temperature anomalies are lower than the PI with  $-0.5 \pm 0.2$  °C. Subsequently, SH  $\Delta$ SSTs exhibits a slow cooling (-0.7 °C) up to  $\sim$ 327 ka , followed by a strong cooling (-0.9 °C) up to  $\sim$ 320 ka. In the NH, this cooling is  $\sim$ 10 ka-long and is larger (-1.9 °C), with the coolest conditions observed at  $\sim$ 318 ka. In the Tropics, a 0.6 °C cooling is observed with a shorter duration ( $\sim$ 7 ka; **Fig. 4D**).

After these cooling periods, the SH and Tropics  $\Delta$ SST are relatively stable up to ~315 ka, while the NH shows a 3 ka- warming phase of ~0.9 °C. Afterwards, the NH  $\Delta$ SST exhibits a cooling (~1.0 °C) in two phases up to 300 ka (**Fig. 4D**). The SH and Tropics exhibit a continuous cooling phase about ~0.9 °C and ~1.1 °C up to 300 ka, respectively.

#### 3.2.3 Basin scale








Our regional  $\Delta$ SST stacks (**Fig. 4E-G**) across the different basins of Atlantic, Pacific and Indian oceans, show similar trends in structure of temperature variability within hemispheres, but differ in terms of timing and amplitude of temperature changes.

In the NH, the MIS 10 glacial estimates indicate cooler conditions in the Atlantic than in the Pacific with values of approximately  $-5.4 \pm 0.3$ °C and  $-3.5 \pm 0.3$ °C respectively (**Fig. 4E-F**). This glacial maximum is attained early in the North Pacific (~349 ka), while the cold peak in the North Atlantic is only attained at ~342 ka. The deglacial in the North Pacific occurs at ~338.5 ka after a ~10 ka-long phase of stability, with a warming of ~3.4 °C over 7 ka. In the North Atlantic, the deglacial warming of 6.3°C occurs at ~342 ka in about 8.5 ka. The  $\Delta$ SST values at their respective climatic optimum, attained at ~333 ka and ~329 ka, are  $1.0 \pm 0.2$ °C and  $0.6 \pm 0.3$  °C (**Fig. 4E-F**) for North Atlantic and Pacific respectively. The shape of the optimum also differs, with a short an abrupt peak in the North Atlantic, directly followed by a cooling, while the  $\Delta$ SST optimum peak in the North Pacific is more smoothed, lasting approximately ~3 ka. The glacial inception in the North Atlantic is characterized by a ~3.8°C cooling over ~15 ka, while it is characterized by a cooling of 1.7°C over 8.5 ka in the North Pacific. The fast warming observed in the NH  $\Delta$ SST stack (**Fig. 4D**) is also well-defined in the North Atlantic, with a 2 ka temperature increase of ~1.1°C (**Fig. 4E**). In the North Pacific, this warming event starts 2 ka earlier than in North Atlantic. It lasts ~4 ka, but the warming amplitude is of ~0.5°C. After this millennial-scale change, the North Atlantic  $\Delta$ SST are quite variable during ~5 ka with several short warming and cooling episodes associated with an amplitude of 0.4 to 0.9 °C until 300 ka (**Fig. 4E**). The North Pacific  $\Delta$ SST exhibits a continuous cooling of 0.8 °C up to 300 ka (**Fig. 4F**).

The Equatorial Pacific  $\Delta$ SST stack exhibits some variations that are similar to those observed in the Tropics  $\Delta$ SST stack described in the previous section. The MIS 10 glacial period extends from 350 (or older) to ~337 ka, with  $\Delta$ SST of -2.3  $\pm$  0.3 °C slowly increasing to -1.7  $\pm$  0.3 °C. The deglaciation is characterized by a ~6.5 ka-long warming of ~2.1 °C (**Fig. 4F**). The interglacial conditions are reached at ~330 ka with a plateau lasting 3.5 ka. From 327 to 300 ka, the  $\Delta$ SST of Equatorial Pacific exhibits a continuous cooling of ~2.0 °C, with a short rebound of temperatures at ~316 ka.

In the SH, the Atlantic and Pacific  $\Delta$ SST stacks show similar long-trend structure of variability, but differ in terms of amplitude and timing of temperature changes (**Fig. 4E-F**). The Indian Ocean  $\Delta$ SST stack (**Fig. 4G**) is different in terms of temporal variability, but similar in terms of amplitude of temperature changes. The South Pacific  $\Delta$ SST stack shows the highest uncertainty of all basins stacks (Fig. 4F). The MIS 10 glacial period is not covered in the South Atlantic stack (Fig. **4E**). However, it extends from ~350 ka to ~342.5 ka in the South Pacific, with a minimum  $\Delta$ SST values of -3.9  $\pm$  0.4 °C (**Fig. 4F**). In the Indian Ocean, the colder values (-3.8  $\pm$  0.3°C) are reached at ~346.5 ka (**Fig. 4G**). The deglacial warming lasts ~12 ka in the South Atlantic (and possibly longer as it is not fully recorded), with a warming amplitude of a 2.7°C (Fig. **4E**). In the South Pacific, the deglaciation is shorter ( $\sim$ 7.5 ka) and with higher warming amplitude of  $\sim$ 3.6°C (**Fig. 4F**). In the Indian Ocean, the deglaciation is characterized by a ~3.4 °C warming from 341 to 330 ka (Fig. 4G). The interglacial conditions are reached at around ~338 ka, ~335 ka and 330 ka in the South Atlantic, Pacific and Indian Ocean, respectively (Fig. 4E-G). The  $\Delta$ SST stacks show warmer conditions relative to PI during the optimum in the South Atlantic (0.2  $\pm$ 0.3°C), the South Pacific (0.2  $\pm$  0.5°C) and the Indian Ocean (0.3  $\pm$  .0.1 °C) (**Fig. 4E-G**). The South Atlantic and Pacific basin  $\triangle$ SST stacks show a cooling phase after this optimum, with durations of ~12.5 ka and cooling amplitudes of ~1.1 and ~2.3°C in South Atlantic and Pacific, respectively (Fig. 4E-F). After this period, short warming phases are observed, with maximum temperatures occurring synchronously between the two basins at ~316 ka. Finally, the two South Atlantic and Pacific stacks show a continuous decrease in temperature up to 300 ka, with cooling amplitudes of respectively ~2.7 and ~1.2°C (**Fig. 4E-F**). In the Indian Ocean, temperature changes after the interglacial peak are marked by a continuous cooling (~1.7 °C) until 300 ka, with a short rebound of temperatures at 305.5 ka (**Fig. 4G**).

#### 4 Discussions







#### 4.1 Limitations associated with our spatio-temporal SST synthesis

#### 4.1.1 Limitations from the individual $\Delta$ SST records

In the Supplementary Information S1, we describe in several instances an "inconsistent signal" between two SST proxies from a single site. These discrepancies, both in terms of variability and absolute values, create an uncertainty on the exact SST variations at a given site. However, when two proxies lead to inconsistent SST reconstructions, it is not always straightforward to determine which one should be preferred. For most of these inconsistent records for a given site, one of the SST values are inferred from the  $\delta^{18}O_p$  proxy. We hypothesize that this discrepancy could arise from the prior  $\delta^{18}O_p$  of

seawater used for the Bayesian calibration (Malevich et al., 2019), based on the  $\delta^{18}O_{sw}$  synthesis from Breitkreuz et al. (2018). In areas where this  $\delta^{18}O_{sw}$  value can drastically change due to various factors such as freshwater from rivers, meltwater and rainfall, it is not surprising to observe a different pattern of SST variability and a biased signal. Without the possibility to define precisely the prior  $\delta^{18}O_{sw}$  for all the sites, we maintain the same procedure for each of them. Additional issues are identified with SST derived from Mg/Ca records. This proxy is a complex paleothermometer, and the Bayesian calibration (Tierney et al., 2019) relies on several uncertain prior that can introduce bias into the records. Missing (or approximative) information, such as the cleaning method used (see Tierney et al., 2019) in original studies can lead to a less accurate calibration of the Mg/Ca ratio. So far, the MAT and  $U_{37}^{K\prime}$  SST reconstructions, which do not require specific external prior (e.g. seawater properties), have not displayed major inconsistencies in SST variability; only SST derived from Mg/Ca and  $\delta^{18}O_p$  exhibit such issues. We therefore propose that differences between SST estimations based on multiple proxies stem from uncertain parameters required in the calibration process. Nevertheless, inconsistent records derived from Mg/Ca or  $\delta^{18}O_p$  proxies are few and are "drowned out" by other SST records during the stacking process. For this reason, we decide not to exclude any of these records.

Another limitation attached to our synthesis is related to the use of a PI SST reference originating from the HadISST database (Rayner et al., 2003). The limitations described above are not relevant for the calibration of the PI SST, as most prior (e.g. seawater properties) are well constrained for the PI period. However, Gao et al. (2024) recently compared the annual SST in the Southern Ocean of the "piControl" experiments from 12 climate models of the Paleoclimate Modelling Intercomparison Project Phase 4 (PMIP4) with the HadISST database for the period 1870–1899. Their findings suggest that in the Southern high latitudes (> 40°S), differences between models and data can locally range from -5 or 5°C. This highlights that uncertainties in the choice of the PI SST can bias the absolute values of anomalies reported here. The use of core-top database could also be considered, since proxy values are directly available. However, these records mainly reflect a "present-day" conditions, which may differ from a PI reference and thereby introduce biases in final global- and regional-scale temperatures estimates. In addition, for core tops it is often unknown how large a time interval is integrated in the individual measurements, which may further limit their suitability as a PI reference. Nonetheless, HadISST remains, to the best of our knowledge, the most representative estimate of monthly SST during the PI period.

## 4.1.2 Limitations from global and regional stacks

In this study, the use of Bayesian approaches and the Monte-Carlo processes, based on the age and ΔSST ensembles, allow us to account for all source of uncertainties (see Section 2.5) and produce continuous time-series SST reconstructions without hiatus. Nevertheless, an important uncertainty remains regarding the spatial distribution of the marine records. The lack of data from the open Pacific Ocean, the central Indian Ocean, the Nordic Seas and the Western Atlantic Ocean may lead to a misinterpretation of latitudinal temperature trends. Such limitation is common when building stacks from data syntheses (e.g. Shakun et al., 2012, 2015; Capron et al., 2014; Snyder, 2016; Hoffman et al., 2017; Tierney

et al., 2020; Osman et al., 2021; Clark et al., 2024) and may affect the final global  $\Delta$ GSST or regional stacks. However, the stacking process used in this study minimize this unavoidable limitation (see **Section 2.5**).

Another challenge is related to the interpretation of the global  $\Delta$ GSST and  $\Delta$ GMST signals. Producing such stacks is very valuable for estimating the global response to natural forcing or comparing global temperature reconstructions between different periods (e.g. Snyder, 2016; Osman et al., 2021; Clark et al., 2024). However, our descriptions of hemispheric (**Section 3.2.2**) and basin scale (**Section 3.2.3**) stacks reveal significant differences in the timing of temperature changes between different regions, particularly during the deglaciation and the interglacial peak. Hence, these asynchronous changes lead to a "smoothed" global response in terms of both temporal dynamics and amplitude of variability. Accordingly, a comprehensive interpretation of temperature changes across T-IV and MIS 9 requires integrating information from individual records, as well as from regional, hemispheric, and global SST stacks.

#### 4.2 Regional temperature changes across MIS 10, T-IV and MIS 9

#### 445 4.2.1 From glacial to interglacial conditions






The deglacial warming during T-IV is marked by major differences in the timing of regional changes (Fig. 5D-G). Notably, the South Atlantic appears as the first regions to warm. This observation aligns with findings by Shakun et al. (2012) regarding the onset of the deglacial warming during the T-I, characterized by a ~2 ka lead in the SH compared to the NH and identified as a result of the bipolar seesaw. This early deglacial warming thus appears to be a consistent feature across glacial terminations and is likely attributed to bipolar seesaw, but may also comes from the progressive export of warm and salty waters from the Agulhas Current (known as "Agulhas Leakage"), which typically flows westward into the south-eastern Atlantic basin during terminations (Peeters et al., 2004; Bard and Rickaby, 2009; Biastoch et al., 2009; Turney and Jones, 2010; Beal et al., 2011; Caley et al., 2011, 2012; Denton et al., 2021; Nuber et al., 2023). In the South Atlantic, a southward shift of westerly winds during a deglaciation (e.g. Gray et al., 2023 for T-I) likely enhanced the Agulhas Leakage, increasing the influx of warm and salty Indian Ocean waters into the South Atlantic (Denton et al., 2021; Nuber et al., 2023). This mechanism is particularly evident during T-IV, where it amplifies SST variability and appears to be the main driver of the early warming observed in the South Atlantic basin. Consequently, we suggest that this early South Atlantic warming reflects an intrinsic oceanic mechanism rather than a direct response to radiative forcing. As most of Southern Hemisphere records are located in the South Atlantic (see **Supplementary Information S1**), this may also explain why SH temperatures begin rising before the increase in CO<sub>2</sub> concentrations and Antarctic air temperatures (Fig. 5A), as also observed during the T-I (Shakun et al., 2012; Osman et al., 2021).

**Figure 5: Comparisons of global and regional temperature anomaly stacks to climate records.** A) δD of the EDC ice-core (grey; Jouzel et al., 2007) and composite  $CO_2$  concentration record (green; Bereiter et al., 2015; Nehrbass-Ahles et al., 2020), B)  $\Delta$ GMST (purple, this study) compared to the GAST from Snyder (2016) (red with  $2\sigma$  uncertainty envelop) and GMST from Clark et al. (2024) (black with  $\sigma$  uncertainty envelop), C) North (blue) and South (red) hemispheric temperature stacks, D) Northern hemisphere minus Southern hemisphere temperature stacks considered as hemispheric heat transfer, E) North (blue) and South (red) Atlantic temperatures stacks, F) North (blue) and South (red) Pacific temperatures stacks obliquity (black), G)  $\delta^{18}O_{calcite}$  composite record (Cheng et al., 2016) rescaled on AICC2023 following Bouchet et al. (2023), H) obliquity and 65°N solstice insolation as anomaly to present and I) IRD count of the ODP site 983 (Barker et al., 2015) rescaled on AICC2023 indicating the iceberg discharges in North Atlantic. Blue areas are related to Heinrich stadial events. Light red area marks the "optimum" of global temperatures as recorded in the  $\Delta$ GMST stack. Shaded envelopes in B to F represent each percentile around the median (solid line) of our new temperature stacks.

Interestingly, North Atlantic and NH temperatures (**Fig. 5C, E**) begin increasing during the Heinrich Event (HE), as recorded by the ice-rafted debris (IRD) in the North Atlantic (**Fig. 5I**; Barker et al., 2015). This pattern mirrors observations from T-I (Shakun et al., 2012). While some North Atlantic sites show significant cooling during the HE and the Henrich Stadial (HS; the cold period accompanying HE) of T-IV (**Fig. S1**), others do not, suggesting heterogeneous impacts of meltwater in the North Atlantic basin. This early warming in the NH may result as a response to the 65°N summer solstice insolation increase (**Fig. 5H**), that would also destabilize the Northern hemisphere ice sheets and trigger the HE of T-IV as recorded in the IRD record (**Fig. 5I**).

In the North Pacific (**Fig. 5F**), late deglacial warming is likely tied to changes in the Asian Monsoon system (e.g. Cheng et al., 2016). The start of warming in the North Pacific stack coincides with the increase in East Asian monsoon rainfall, as recorded in  $\delta^{18}O_{\text{calcite}}$  from Sanbao Cave (**Fig. 5G**; Cheng et al., 2016). Thus, the deglacial warming pattern observed in the North Pacific may be a response to the same mechanism affecting the East Asian monsoon, which is most likely related to the Northern high latitude insolation (Cheng et al., 2016). The early warming in the South Pacific compared to the North Pacific would be attributed to the bipolar seesaw mechanism (Stocker & Johnsen, 2003). Its deglacial warming is aligned to the Antarctica air temperature and  $CO_2$  level rises (Jouzel et al., 2007; Bereiter et al., 2015; Nehrbass-Ahles et al., 2020; Fig. 6A), suggesting that radiative forcing could be the main forcing in the South Pacific.

We have highlighted that regional SST in the Atlantic Ocean are influenced by mechanisms associated with two important climate features, the Agulhas Leakage and HE. These events also impact the Atlantic Meridional Overturning Circulation (AMOC), either by weakening it during HE (e.g. McManus et al., 2004; Henry et al., 2016) or strengthening it through salt-water influx from the Agulhas Leakage (e.g. Beal et al., 2011; Caley et al., 2012; Nuber et al., 2023). These SST changes in the Atlantic Ocean can modify AMOC strength, thereby altering hemispheric heat transfer (Shakun et al., 2012). To illustrate this, we calculated an additional stack representing hemispheric heat transfer by subtracting Southern temperature anomalies to Northern ones (**Fig. 5D**). During the HE, the heat transfer is low, reflecting a weakened AMOC. Reduced heat transfer to the North Atlantic likely shifts atmospheric cells southward, leading to a weakened Asian monsoon (e.g. Wassenburg et al., 2021) and persistent cold conditions in the North-west Pacific (**Fig. 5F**). At ~343 and ~337 ka, the heat transfer successively increases, suggesting an AMOC recovery despite ongoing HE conditions in the North Atlantic (**Fig. 5D**). Climate simulations by Nuber et al. (2023) confirm that salt influx in the South Atlantic during HE can drive

AMOC recovery. Thus, AMOC dynamics, regulated by salt and heat fluxes (that affect density gradient between low and high latitudes; H. Stommel, 1961), emerge as a primary driver of regional deglacial temperature patterns.

To summarize, the temperature variability during the T-IV is primarily driven by radiative forcing such as insolation or  $CO_2$  concentration increases. However, the differences observed between the different basins may be related to internal forcing such as Agulhas Leakage, massive iceberg discharges (HE), AMOC dynamics changes or atmospheric reorganization (as illustrated by changes in East Asian monsoon).

#### 4.2.2 From interglacial optimum to the glacial inception







Regional stacks reveal a highly heterogeneous interglacial optimum during MIS 9e, differing in timing, ASST values, and temporal patterns. As mentioned, early South Atlantic warming during the interglacial peak likely results from increased Agulhas Leakage (Peeters et al., 2004; Bard and Rickaby, 2009; Biastoch et al., 2009; Turney and Jones, 2010; Beal et al., 2011; Caley et al., 2011, 2012; Denton et al., 2021; Nuber et al., 2023). This warming is soon followed by cooling as heat is transferred northward via a strengthened AMOC (Fig. 5D). The onset of the "plateau" in hemispheric heat transfer (~333 ka) coincides with the North Atlantic temperature peak, underscoring the role of internal oceanic dynamics in shaping Atlantic SST variability. The subsequent cooling in both hemispheres occurs as hemispheric heat transfer stabilizes (i.e. strong AMOC), with a larger cooling in SH. This suggest that forcing external to ocean dynamics (e.g. Insolation) becomes the dominant control over SST variability in SH (Fig. 5A, H), while NH is still affected by regional processes (e.g. Asian monsoon, AMOC). At the end of the cooling phase of the glacial inception, hemispheric heat transfer decreases, suggesting a slowdown of ocean dynamics. At the same time (317-320 ka; Fig. 5), a slow warming occurs in the South Atlantic while a pronounced cooling in the North Atlantic is observed (Fig. 5E). This bipolar seesaw millennial-scale event aligns with a typical HS SST response, as also indicated by IRD counts in the North Atlantic (Barker et al., 2015; Fig. 51). This seesaw pattern underscores AMOC's critical role in redistributing heat (e.g. Stocker and Johnsen, 2003; J. Lynch-Stieglitz, 2017; Pedro et al., 2018; Davtian and Bard, 2023). The final temperature warming shift in North Atlantic at ~317 ka coincides with reduced IRD count, indicating the end of the HE.

In the South Pacific, the interglacial peak is likely in phase with the one observed in surface air temperature recorded at EDC (**Fig. 5A**; Jouzel et al., 2007). The continuous cooling after this peak also follow Antarctic signals up to 320 ka (**Fig. 5F**). This peak leads the interglacial plateau observed in North Pacific by  $\sim$ 4 ka (**Fig. 5F**). Therefore, in the South Pacific, radiative forcing as insolation or GHG may be the major feature that drives the SST changes. However, we do not exclude that the Antarctic Circumpolar Current (ACC) strength during MIS 9e (e.g. Lamy et al., 2024) could partially influence South Pacific SST variability. In the North Pacific, the  $\Delta$ SST variability during interglacial and glacial inception likely resemble to the Asian Monsoon variability, suggesting that local variability may be a direct response to the same forcing, that is likely northern summer insolation (Cheng et al., 2016).

Hemispheric  $\Delta SST$  during the optimum exhibit a significant warming in extra-tropic hemispheres, with mean temperatures as warm or warmer than the PI period, while the inter-tropic areas appears to be colder (**Fig. 4D**). This

observation was also made for the LIG (Hoffman et al., 2017) and is related to polar amplification (e.g. Holland and Bitz, 2003; Masson-Delmotte et al., 2010). These differences between low and high latitude temperatures are related to a larger amount of insolation forcing received in high latitudes during the optimum (**Fig. 4A**) and albedo and sea-ice positive feedbacks (Capron et al., 2017). A larger obliquity (**Fig. 5H**) can also be responsible of more contrasts in latitudinal solar energy received and reduces the sea-ice areas (e.g. Yin et al., 2021).

To summarize, the MIS 9e optimum is also primarily affected by radiative forcing and surface ocean dynamics. However, the quasi-absence of AMOC changes, freshwater fluxes or changes in atmospheric circulation suggest that MIS 9e temperature variability is primarily shaped by radiative forcing with less influence of regional changes compared to T-IV.

#### 4.3 New estimates of GMST and comparisons to previous estimates







Building on the regional patterns described above, our new  $\Delta GMST$  stack offers an opportunity to refine previous estimates of  $\Delta GMST$  during T-IV and MIS 9. The  $\Delta GMST$  integrates temporal, timing, and amplitude variations across different regional stacks.

Our new estimate of  $\Delta$ GMST provides a new perspective for studying MIS 9 and T-IV climatic response to natural forcing mechanisms (i.e. without anthropogenic forcing). As shown in **Fig. 5B**, the new  $\Delta$ GMST is compared with previous global averages such as the Global Average Surface Temperature (GAST; Snyder, 2016) and Global Mean Surface Temperature (GMST; Clark et al., 2024). In terms of temporal variability, the three datasets exhibit broadly similar trends. However, a significant short-term warming event around ~317 ka (**Fig. 5B**), also recorded in the GSST stack Shakun et al. (2015), is not recorded in the GAST estimate (Snyder, 2016) and is less pronounced in the GMST (Clark et al., 2024). Therefore, our high-resolution SST synthesis made with careful attention on chronologies allows to better detect millennial-scale variability and refine the chronological framework of temperature changes over this period of time.

During the MIS 10 glacial and the cooling phase at ~305 ka, the three Global temperature estimates converge within a similar range of anomalies relative to the PI. However, the GMST from Clark et al. (2024) is ~1°C warmer during glacial conditions compared to our  $\Delta$ GMST (**Fig. 5B**). The deglacial warming (from minimal to maximum temperatures) in our  $\Delta$ GMST (~5.7 °C) aligns with Clark et al. (2024) (~5.2°C) and Snyder estimate (~5.9°C; Snyder, 2016). During the interglacial optimum, the previous stacks exhibit global temperature similar between them (from 0 to 0.3 relative to PI), while our new estimate suggests a slightly cooler interglacial peak of -0.3 °C  $\pm$  0.3°C (**Fig. 5B**). The subsequent cooling phase shows the most notable differences between our new estimate and the previous ones, where our  $\Delta$ GMST is often ~1.0°C cooler (**Fig. 5B**). While our values fall within the 2 $\sigma$  uncertainty of the GAST from Snyder (2016) during this period, they are outside of the lower range of the GMST uncertainties in Clark et al. (2024).

The existing MIS 9e Mean Ocean Temperature (MOT) estimate of ~2°C above the Holocene average (Shakun et al., 2015; Haeberli et al., 2021) exceeds our  $\Delta$ GSST estimate by ~2.1°C (**Fig. 4C**), a difference twice as large as that observed for the LIG (Hoffman et al., 2017; Shackleton et al., 2020). The MOT is largely determined by high-latitude regions, where deep or

intermediate waters are formed, and is homogenized by meridional overturning circulation (Shackleton et al., 2020). Therefore, polar SST records are essential to compare SST and MOT estimations. Unfortunately, no published SST records with sufficient temporal resolution are available for MIS 9e in Nordic Seas. Another potential factor behind these discrepancies could be linked to ocean dynamics. Indeed, a slowdown of AMOC may have resulted in increased heat storage in deep and intermediate ocean (Shackleton et al., 2020; Haeberli et al., 2021). Recent reconstruction of deep-water current strength (Stevenard et al., 2024) and Atlantic  $\delta^{13}$ C synthesis (Bouttes et al., 2023), however, do not show drastic change in deep circulation during MIS 9e compared to other interglacial, excluding a heat storage related to a weakened AMOC. Therefore, deep ocean changes cannot explain such differences between  $\Delta$ GSST and MOT, and this discrepancy may come from the lack of SST data in key areas, as Nordic Seas where deep water are formed, or the Indian Ocean, one of the world's largest heat absorbers during an AMOC "collapse" (Pedro et al., 2018).

Figure 6: Comparison of our new MIS 9 synthesis to the Holocene and MIS 5e ones. A) 65°N summer solstice insolation (light orange, Laskar et al., 2004) and obliquity (dark purple, Berger & Loutre, 1991) variations; B) composite CO<sub>2</sub> (brown; Bereiter et al., 2015; Nehrbass-Ahles et al., 2022) and CH<sub>4</sub> (green, Loulergue et al., 2008) atmospheric concentrations; C) anomaly of Antarctic surface air temperature (light blue; Landais et al., 2021) and 2-ka moving average (blue); most recent GMST for the Holocene and T-I (Osman et al., 2021; Clark et al., 2024), the MIS 5e (Hoffman et al., 2017; Clark et al., 2024) and MIS 9e (this study; Clark et al., 2024). Note that CO<sub>2</sub>, CH<sub>4</sub> and ΔT<sub>site</sub> are plotted on the AICC2023 chronology (Bouchet et al., 2023). We rescale the global SST from Hoffman et al. (2017) on AICC2023 and convert it into GMST by applying the same procedure as in this study. The Osman et al. (2021) stack refers as the "data-only" rather than the estimations data-assimilated into climate model simulations. Red vertical bands are related to deglacial warming periods, yellow vertical bands to the climatic optima (visually defined) and blue vertical bands to glacial inceptions.

# 4.4 Comparing our temperature synthesis over T-IV and MIS 9e with existing syntheses over other deglaciations and interglacials








The comparison to other interglacial SST syntheses (Hoffman et al., 2017; Osman et al., 2021; Clark et al., 2024; **Fig. 6**) suggests that the warming amplitude (~5.7°C) and duration (~ 10.5 ka) of the T-IV are similar to those observed for T-I and T-II (**Fig. 6**). Intriguingly, the glacial maximum during T-II is ~2°C warmer (Clark et al., 2024) than the similar climate state observed during T-I (Osman et al., 2021) and T-IV (**Fig. 6D**).

The interglacial maximum temperature is reached approximately 2 or 3 ka after the GHG overshoot and Antarctic temperature peak, which is also observed in other past interglacials. Global mean temperatures during the optimum of MIS 9e are slightly cooler than PI conditions, the early Holocene (Osman et al., 2021) and the LIG peak (Hoffman et al., 2017; Fig. 6). Compared to the Holocene and MIS 5e interglacial peaks, MIS 9e exhibits a "stable" phase, whereas the others show a continuous cooling trend leading to glacial inception. Osman et al. (2021) demonstrated that this Holocene cooling phase arises from a bias in data synthesis (i.e. influenced by the geographical distribution of data), whereas assimilating these data into climate model simulations provides a more accurate estimate of global temperature changes.

In terms of orbital forcing, the MIS 9e and Holocene periods (Osman et al., 2021; Clark et al., 2024) show temperature optima occurring 2 and 3 ka after the insolation peak, respectively, whereas the LIG temperature peak coincides with the summer solstice insolation maximum (**Fig. 6A, D**). We hypothesize a link with obliquity, which affects latitudinal SST contrasts between low and high latitudes, potentially impacting AMOC strength (e.g. Zhang et al., 2017; Yin et al., 2021), and thereby heat transfer. Moreover, contrasts in latitudinal insolation (**Fig. 4A**) result in a late optimum in low latitude areas, pulling the overall global temperatures towards younger ages. The apparent synchronicity between GMST and insolation changes during the LIG (Hoffman et al., 2017; Clark et al., 2024) may be associated with the early obliquity peak, which likely reduced latitudinal contrasts during the insolation maximum.

Interestingly, the global temperature response during these three interglacial peaks is not proportional to radiative forcing, with MIS 9e exhibiting a muted response compared to the Holocene despite receiving higher insolation energy. As described in previous sections, the asynchronous regional timing of interglacial peaks results in a "smoothed" global response. The differences between the Holocene and MIS 9e can be explained by shorter regional temperature peaks during MIS 9e (**Fig. 5**), while the Holocene exhibits relatively stable hemispheric evolution (Osman et al., 2021). Thus, given the differences in temporal evolution, to better understand climate processes and feedback during past interglacials, it is crucial to look into the regional patterns of changes rather than the global-scale temperature variability.

MIS 9e provides a unique context to study carbon cycle forcing, as it exhibits the highest CO<sub>2</sub> and CH<sub>4</sub> concentrations of the last 800 ka (Loulergue et al., 2008; Nehrbass-Ahles et al., 2022). The increase in atmospheric CO<sub>2</sub> starts just before the initial deglacial warming, which is mainly modulated by the early warming in the South Atlantic. Gray et al. (2023) demonstrated that a southward shift of westerlies accompanies the atmospheric CO<sub>2</sub> increase during T-I, mainly by increasing the carbon release *via* upwelling in the Southern Ocean. Upwelling of deep, carbon-rich waters due to shifts in wind patterns or ocean circulation changes may have released substantial CO<sub>2</sub> into the atmosphere, amplifying warming

trends (Sigman et al., 2010; Gray et al., 2023). This mechanism is likely the same for T-I and T-IV (and probably T-II), demonstrating the complexity of regional interactions between the different climate spheres. Interestingly, while the  $CO_2$  overshoot during MIS 9e is larger in magnitude than that of MIS 5e, the global mean temperature response appears muted in our  $\Delta$ GMST reconstructions of both MIS 5e and MIS 9e. This discrepancy may reflect differences in temporal phasing between interglacial peaks in different regions. However, as discussed in previous sections, even at the regional scale, no major local temperature peaks can be linked directly to this significant atmospheric carbon overshoot. This raises questions about the direct influence of a rapid "jump" in atmospheric  $CO_2$  (Nehrbass-Ahles et al., 2022; Legrain et al., 2024) on regional or global temperatures, particularly given the fast subsequent decline to interglacial levels. Another explanation may be related to the millennial-scale resolution records used in these syntheses, which are unable to record the SST response to these  $CO_2$  jumps. At present, climate model simulations for MIS 9e remain limited, making it difficult to disentangle the causes and effects of this carbon overshoot. Future simulations incorporating ocean-atmosphere carbon exchanges would offer critical insights into the carbon cycle role during MIS 9e. Such models could help elucidate how AMOC variations and Southern Ocean processes influenced  $CO_2$  and shaped global climate responses.

#### 4.5 Warming rate during T-IV compared to modern warming

The Termination T-IV was described as the deglaciation showing the most extreme sea-level rise over the last 800 ka (Grant et al., 2014). However, as mentioned in the previous sections, the amplitude of warming, estimated at  $\sim$ 5.7°C, is similar to those observed during T-I and T-II (**Fig. 6D**) and the MIS 9e interglacial temperature peak appears to be cooler than during the early Holocene (Osman et al., 2021) and the LIG (Hoffman et al., 2017; Clark et al., 2024).

To further examine the deglacial warming trend, we compare the warming rate per century across these three terminations using the first derivative of GMST data (**Fig. 7**). These warming rates were obtained using randomly drawn GMST values (within the  $2\sigma$  uncertainty for each time-bin) in a Monte-Carlo ( $10^5$  iterations) process for each Termination. This comparison indicates that the mean rate of warming is similar for T-II and T-IV ( $0.03^{\circ}$ C per century) but slightly higher for T-I ( $0.05^{\circ}$ C per century). However, the distribution of observed warming rates is skewed toward higher values for T-I ( $99^{th}$  percentile ~ $1.1^{\circ}$ C per century) and T-IV ( $99^{th}$  percentile ~ $0.5^{\circ}$ C per century) compared to T-II ( $99^{th}$  percentile ~ $0.2^{\circ}$ C per century). We suspect that differences in the resolution of GMST reconstructions contribute to these extreme warming rates observed, underscoring the importance of producing high-resolution SST syntheses to better constrain the rates of change (**Fig. 7**).

Deglaciations represent the most extreme natural climatic changes of the Pleistocene period (e.g. Broecker and Denton, 1990; Cheng et al., 2009; Denton et al., 2010; Cheng et al., 2016). However, present-day warming, as illustrated by the HadCRUT5 dataset (Morice et al., 2021), stands out as one of the fastest warming periods (Calvin et al., 2023). To contextualize current warming trends in the framework of past climate extremes, we calculated different warming rates from HadCRUT5 over the last 60 to 170 years (before 2023; **Fig. 7**). Notably, for T-I (Osman et al., 2021), which has the highest-resolution GMST stack, the temperature increase over the last 60 years (Morice et al., 2021) is almost twice as high as the

99<sup>th</sup> percentile of T-I warming rates. For T-IV, the 99<sup>th</sup> percentile (~0.5°C per century) is almost three times lower than the current warming trend observed over the last 80 years. This comparison starkly highlights the exceptional nature of modern warming, which significantly outpaces the largest natural temperature changes recorded over the last 400 ka.

**Figure 7: Percentages of rate of warming over termination I, II and IV.** A) Percentages per bin of warming rate per century for the T-I, T-II and T-IV; B) the HadCRUT5 (Morice et al., 2021) temperature curve, representative of the most recent (1850 to 2023 years CE) GMST. These rates were obtained using the first derivative with  $10^5$  iterations, which randomly drawn the GMST values within the  $2\sigma$  uncertainty for each time-bin. Periods involved for the terminations are 20 to 10 ka (T-I, grey), 140 to 125 ka (T-II, orange) and 346 to 333.5 ka (T-IV, blue). Coloured stars represent the  $99^{th}$  percentile observed for each deglaciation. We calculate the warming trend, from the last 170 to the last 60 years (20 years step). Then, after a conversion in °C per century, we report these trends in the principal panel (vertical bands) for a comparison to those obtained for the three terminations.

#### 5 Conclusions and perspectives




This study provides the most comprehensive SST synthesis covering MIS 9e and T-IV to date. It includes 98 marine sediment SST reconstructions and it benefits from a consistent dating between the different records as well as quantitative estimates of all source of uncertainties associated with both the chronologies and with the use of SST proxies. Hence, this compilation provides the first insights on the deglacial and interglacial surface temperature variability at both global and

regional scales over this time period characterized by the highest natural CO<sub>2</sub> concentrations over the past 800 ka. Our results highlights the following points:

- Hemispheric temperature stacks reveal pronounced asynchrony both in the timing and magnitude of deglacial warming across regions. The extra-tropics bands are characterised by warmer optimum (~0.1 and ~0.2°C for Northern and Southern hemispheres, respectively) than intra-tropic areas (~-0.5°C), suggesting a strong polar amplification of warming. A late optimum is also observed in intra-tropic areas, more affected by external forcing than extra-tropics.
- The regional SST patterns appear to be affected by global radiative forcing at first order, but are mainly influenced by regional features as water exchanges between Atlantic and Indian ocean, meltwater discharges or changes in atmospheric and oceanic dynamics.




- We provide refined estimates of T-IV and MIS 9e  $\Delta$ GMST, relying on our new synthesis that is based on a larger number of records associated with a higher temporal resolution and a more detailed chronological framework than the published compilation covering this time interval. Hence, the deglacial warming during T-IV is of ~5.7°C and reaches a maximum value of -0.3  $\pm$  0.3°C relative to PI during the MIS 9e. Such subdued global temperature estimate results from the temporal asynchronicity of the more pronounced regional temperature changes.
- The ΔGMST reconstruction highlights that the global deglacial warming amplitude during T-IV (~5.7 °C) is comparable to the ones observed during of T-I and T-II, but MIS 9e remains cooler than the early Holocene and LIG. This muted response contrasts with the high CO<sub>2</sub> and CH<sub>4</sub> concentrations of MIS 9e peak, underscoring the need to better understand feedback mechanisms governing global and regional temperature responses.
- Despite differences in orbital configurations and GHG forcing, past interglacials exhibit recurrent patterns, such as
  delayed global temperature peaks relative to GHG and Antarctic temperature changes. However, while the
  Holocene and MIS 5e temperatures indicate a cooling after reaching the optimum, MIS 9e stands out and show a
  relatively stable plateau before the glacial inception.
- Comparisons of past warming rates reveal that T-IV and T-II share similar mean warming rates (~0.03°C per century), with higher extremes observed in T-I. We found that high-resolution GMST leads to a better estimation of "extreme" warming rates during deglaciations. Present-day warming rates over the last century are exceptional, far exceeding even the 99<sup>th</sup> percentile of natural variability during these three past terminations.
- Our new results call for several future research directions to better understand mechanisms involved in past terminations and interglacials periods:
  - In several instances, we found inconsistent signals from a single site with different SST estimates derived from different proxies. Despite the recent improvements in SST calibrations, the lack of prior parameters (e.g. seawater properties) needed for calibrations leads to these inconsistent temporal variabilities or absolute SST values. Efforts should be made to produce more estimations of past seawater properties data to better constrain the SST calibrations.

- High-resolution SST records from polar and subpolar regions are essential to resolve the spatial variability and
  constrain the role of high-latitude processes in interglacial climates. Additionally, integrating paleo-proxies for
  AMOC dynamics would address key uncertainties about its role in modulating heat transport and carbon cycles.
- Comparing the MIS 9e ΔGMST to existing MIS 5e and Holocene GMST provides insights on our understanding of past climate response during warm intervals associated with slightly different forcing. Additional syntheses should cover more interglacials and Terminations to progress on our understanding of the dynamics and the diversity of the Quaternary interglacials.
  - This new compilation is a useful benchmark to evaluate ESM simulations that have been performed over this time interval. Such data-model comparisons would enable to disentangle the interplay between orbital forcing, GHG variations or ocean dynamics over this period.
  - Data assimilation offers a transformational approach to paleoclimate studies by integrating disparate proxy datasets
    with climate model outputs, enabling temporally and spatially complete reconstructions. Its application with our
    new MIS 9 synthesis could reduce biases arising from uneven proxy coverage and improve the determination of the
    global and regional climate patterns.

Finally, T-IV and MIS 9e illustrate the complexity of natural climate transitions, driven by a combination of orbital dynamics, GHG forcing, and ocean-atmosphere feedbacks. While these processes produced significant climatic shifts over millennia, their magnitude and pace are minimal compared to the current anthropogenic warming. By contextualizing modern warming within the late Quaternary framework of Earth's climatic history, this study reaffirms the extraordinary nature of the current anthropogenic warming and the urgent need for decisive action to mitigate its impacts.

# Acknowledgements






This study is an outcome of the Make Our Planet Great Again HOTCLIM project; it received the financial support from the French National Research Agency under the 'Programme d'Investissements d'Avenir' (ANR-19-MPGA-0001). We thank editor B. Risebrobakken and two anonymous reviewers for fruitful discussions that help improving the manuscript. We thank all authors who kindly shared their SST datasets with us, *via* personal communication or by the way of international FAIR database. We also thank Frédérique Parrenin for fruitful discussions about the chronological aspect of this study.

#### **Data availability**

All references of data used in this synthesis are listed in **Table S2**. The revised age models, SST, tie-points and global- and regional-scale stacks are available in a Zenodo repository (<a href="https://doi.org/10.5281/zenodo.17174964">https://doi.org/10.5281/zenodo.17174964</a>).

#### 730 Authors contributions

NS and EC designed the research. NS collected the datasets building on a preliminary effort undertaken by CC. NS computed age models, SST calibrations and developed the method to infer the global and regional temperature stacks. NS lead the writing of the manuscript with subsequent inputs from EC and EL.

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
