# Peer review of "Global and regional sea-surface temperature changes over the Marine Isotopic Stage 9e and Termination IV"

_EGUsphere, 2025_

## Author Comment (AC2)

The study by Stevenard et al. presents the first high-resolution synthesis of sea surface temperatures (SST) across the final portion of Marine Isotope Stage 10 (MIS 10), Termination IV and MIS 9, incorporating 98 carefully recalibrated and re-dated marine records. By applying a consistent chronological framework (AICC2023), updated SST calibrations, and a Monte Carlo approach to quantify uncertainties from both dating and SST reconstructions, the authors generate global and regional SST stacks that reveal multi-millennial climate dynamics and discuss their potential drivers.

This work is highly relevant in the context of understanding warm climate intervals, particularly as MIS 9e is associated with the highest atmospheric $CO_2$ and $CH_4$ concentrations of the past 800 kyr. The study is of significant value to the paleoclimate community, as these SST stacks provide a basis for evaluating the global climate response to natural forcings (global and regional perspectives) and allow meaningful comparisons with other interglacial periods. Furthermore, the authors are transparent about the limitations of their approach and provide thoughtful perspectives on how future work can build upon these findings.

Importantly, the study offers refined estimates of deglacial mean surface temperature (DGMST) for the target period, based on a larger and higher-resolution dataset than previous compilations.

The manuscript is scientifically sound, methodologically robust, and well-structured. Overall, I find the study to be a valuable contribution and worthy of publication following minor revisions as outlined below.

**Specific comments**

Abstract: homogenize the use of "MIS 9" and "MIS 9e" throughout the abstract for consistency.

We have homogenized the use of MIS 9e throughout the abstract.

The introduction could benefit from improved flow and narrative cohesion. Consider reorganizing some paragraphs or refining transitions between concepts to enhance clarity and better guide the reader through the motivation, context, and objectives of the study.

We have refined transitions between concepts to enhance clarity and better guide the reader.

*"[...] rapid sea-level rise. Yet, despite this relevance, our understanding of global or regional temperatures during this period still relies mostly on reconstructions derived from long-term temperature stacks (e.g. Shakun et al., 2015; Snyder, 2016; Friedrich et al., 2016; Clark et al., 2024). For [...]" — Lines 58-59.*

*"[...] the preceeding Termination IV. Beyond chronological uncertainties, the variety of SST proxy and the improvement in proxy-calibrations over decades challenge a direct comparison between published records. The existing SST [...]" — Lines 76-77.*

Lines 76–79: Please provide references for the mentioned sea surface temperature proxies during the target interval.

As described in Table S1, many references can be used to illustrate SST records covering MIS 9e. Rather than adding specific references, we refer the reader to Table S1, which lists all citations. —Line 78.

*"[...] a wide range of proxies (see Table S1), [...]"*

Table S1: Consider providing more detailed information about the proxies used to estimate SST. For example:

- Specify the foraminifera species used in the Mg/Ca and $\delta^{18}O_P$ records to clarify whether the signal reflects seasonal or annual conditions.
- Consider also indicating the ocean basin or region from which each record originates, even though latitude and longitude are already provided.

We have added two columns "Basin" and "Species" in the Table S1.

Lines 101-103: a more detailed description of the used proxies is needed in the main text. For example:

- How each proxy works
- The type of signal it reflects (seasonal vs. annual)
- Why proxies with seasonal and annual signals are considered, comparable and suitable for joint inclusion in the compilation

We have revised this paragraph and include it in the introduction. Briefly, we have added a more detailed description of the existing SST proxies: how they work and the expected type of SST (annual or seasonal). —Lines 79-95.

*"The alkenone unsaturation index ($U_{37}^{K'}$) is based on the relative unsaturation of long-chain (C37) alkenones, which decreases with increasing temperature (Prahl et al., 1988). It typically reflects annual SST, but may capture seasonal temperatures at high latitudes (~45°N) or in semi-enclosed basins such as the Mediterranean (Tierney and Tingley, 2018). The Modern Analogue Technique (MAT) estimates SST by comparing fossil planktic assemblages (e.g. foraminifera, radiolarians) with modern datasets using a transfer function, identifying the closest analogues based on species compositions (e.g. Ruddiman et al., 1989). MAT generally reflects seasonal SST. The magnesium-to-calcium (Mg/Ca) ratio of planktic foraminifera reflects a nonlinear increase in magnesium incorporation into calcite shells with rising temperature (Oomori et al., 1987). While often associated with mean annual SST, it may reflect seasonal conditions depending on the foraminifera species and the site's latitude (Tierney et al., 2019). The $\delta^{18}O$ from planktic foraminifera ($\delta^{18}O_p$) is affected by both temperature and the isotopic composition of seawater ($\delta^{18}O_{sw}$) at the time of calcification. Although calibration is complex due to varying relationships derived from culture and plankton tow studies, recent Bayesian approaches (Malevich et al., 2019) allow for SST estimates that incorporate uncertainties in $\delta^{18}O_{sw}$ and can reflect either annual or seasonal SST, depending on species and calcification latitude (Malevich et al., 2019). The $TEX_{86}$ (TetraEther indeX of 86 carbons) proxy is based on the relative distribution of archaeal glycerol dibiphytanyl glycerol tetraether (CDGT) lipids produced by marine archaea (Schouten et al 2002). It can estimate SST through a temperature-dependent change in ring structures, but often reflects subsurface conditions (e.g. Schouten et al 2002; Lopes Dos Santos et al., 2010; Tierney & Tingley, 2014; Rouyer-Denimal et al., 2023), limiting its comparison with surface-based SST reconstructions."*

We have added a small sentence at the end of the introduction about the choice of keeping both annual and seasonal signals.

*"[...] records. To maximize spatial and temporal coverage, we include both annual and seasonal SST records, while primarily interpreting the results in terms of annual mean temperatures. To ensure [...]"*

Line 525: The title of Section 4.1 could better reflect its content by including a reference to Termination IV and MIS 10. For example: *"Limitations associated with our MIS 10–TIV–MIS 9 spatio-temporal SST synthesis"* or simply *"Limitations associated with our spatio-temporal SST synthesis."* This logic could also be applied to the titles of other manuscript sections for improved clarity.

We have modified the title of section 4.1 to "Limitation associated with our spatio-temporal SST synthesis". —Line 391.

Lines 604–608: You describe delayed SST warming in the North Pacific (relative to the North Atlantic and Northern Hemisphere stacks) and compare it to Asian monsoon and insolation records. However, as with the Atlantic, the South Pacific appears to have warmed earlier than the North Pacific. For the Atlantic, you suggest this interhemispheric difference may be linked to intensified Agulhas Leakage during deglaciation. A brief discussion of possible mechanisms explaining the earlier South Pacific warming is currently missing.

We have added a short paragraph to discuss about possible mechanisms explaining the early South Pacific warming after the discussion about the North Pacific. —Lines 479-482.

*"[...] high latitude insolation (Cheng et al., 2016). The early warming in the South Pacific compared to the North Pacific would be attributed to the bipolar seesaw mechanism (Stocker & Johnsen, 2003). Its deglacial warming is aligned to the Antarctica air temperature and $CO_2$ level rises (Jouzel et al., 2007; Bereiter et al., 2015; Nehrbass-Ahles et al., 2020; Fig. 6A), suggesting that radiative forcing could be the main forcing in the South Pacific."*

Line 632: Regarding "(~333.5 ka)": consider providing an age interval for the hemispheric heat transfer plateau, rather than a single date. Alternatively, clarify that ~333.5 ka marks the onset of the plateau.

We have clarified and revised the dates of the Hemispheric Heat Transfer records throughout the text. —Line 505.

*"[...] strengthened AMOC (Fig. 5D). The onset of the "plateau" in hemispheric [...]"*

Line 638: It seems that the most appropriate term in this context is Heinrich Stadial. Since Heinrich Stadial (HS) and Heinrich Event (HE) have distinct definitions, please double-check the use of the term "HE" here and throughout the manuscript to ensure consistency and accuracy.

We have modified the use of HE and named the appropriate period HS throughout the text (also in Figure 6; now Fig. 5 in the revised manuscript).

Line 642: The phrase "In contrast, Pacific DSST changes are less abrupt" is vague. Less abrupt than what? The North Atlantic DSST? The Atlantic as a whole? Please specify.

We have removed this sentence as the new stacks estimates give another point of view.

Lines 656–657: A reference to Figure 6H would be appropriate here.

We have added a reference to the figure 6H in the sentence (now figure 5H). —Line 530.

*"A larger obliquity (Fig. 5H) can [...]"*

Lines 688–689: *"Indeed, a slowdown of AMOC may have resulted in increased heat storage"* — please clarify: increased heat storage where? In the Southern Ocean? In the deep ocean?

We have clarified this point at the end on the sentence. —Line 563.

*"Indeed, a slowdown of AMOC may have resulted in increased heat storage in deep and intermediate ocean (Shackleton et al., 2020; Haeberli et al., 2021)."*

**Technical corrections**

1. When providing time intervals, please list the older date first (i.e., from past to present). For example, use "335 to 300 ka" instead of "300 to 335 ka".

   We have "reversed" all time interval throughout the manuscript.

2. Line 12: Correct to "…from the fact that **it** is associated…"

   We have modified this sentence. —Line 12.

   *"[...] from the fact that it is associated with the highest atmospheric $CO_2$ concentrations [...]"*

3. Line 14: Correct to "…of sea surface temperatures (SST) are available…"

   We have modified this sentence. —Line 14.

   *"Numerous reconstructions of sea surface temperatures (SST) cover this time interval [...]"*

4. Line 41: Correct to "sometimes exhibiting conditions as warm as…"

   We have modified this sentence. —Line 40.

   *"[...] sometimes exhibiting conditions as warm or [...]"*

5. Line 193: Typo — correct to "planktic foraminifera".

   We have modified this sentence. —Line 213.

   *"[...] (i.e. $\delta^{18}O$ of planktic foraminifera after [...]"*

6. Line 200: Typo — correct to "Three scenarios".

   We have modified this sentence. —Line 221.

   *" Three scenarios occur: (1) the site [...]"*

7. Line 447: Typo — "(i.e. when the global temperature starts to rise sharply)".

   We have modified this sentence. —Line 312.

   *"[...] (i.e. when the global temperature starts to rise sharply)."*

8. Figure 5D: The behaviour of the tropical stack during MIS 10 is not clearly visible. Please also check the other figures.

We have modified the figure (now Fig. 4 in the revised manuscript) and the others by putting the median (solid line) of each stack in the foreground.

9. Lines 575–576: Typo — "characterized by a ~2 ka lead in the SH compared to the NH".

We have modified this sentence. —Lines 443-444.

*"[...] characterized by a ~2 ka lead in the SH compared to the NH and identified [...]"*

10. Lines 598 and 603: Standardize the formatting of "Heinrich event" throughout the manuscript (choose either "Heinrich Event" or "Heinrich event" and apply consistently).

We have standardized the "Heinrich Event" expression throughout the manuscript.

11. Figure 6: Double-check the y-axis title for panel D.

The y-axis title for panel D is "Hemispheric heat transfer" as desired by the authors. Note that after the submission, we have added a commentary with the new figure without errors.

12. Line 615: Update citation to refer to Figure 6D instead of 6E.

We have modified the citation (now Fig. 5D in the revised manuscript). —Line 489.

*"[...] to Northern ones (Fig. 5D)."*

13. Standardize the terminology for "greenhouse gases" throughout the manuscript (e.g., use consistently: "GreenHouse gases", "GHG", or "greenhouse gas").

We have standardized by using "GHG" (once defined).

---

## Author Comment (AC3)

**General comments**

Overall, this is a well-written and thoughtful paper. Below, I briefly address the criteria outlined by the journal: the manuscript tackles relevant scientific questions within the scope of the journal and presents novel approaches that contribute to the field. The authors reach substantial conclusions, which are generally supported by the results. However, the scientific methods and underlying assumptions require further clarification, particularly regarding experimental procedures and calculations, to ensure reproducibility. The authors appropriately acknowledge related work and clearly delineate their original contributions. The title and abstract accurately reflect the content of the paper. While the language is mostly fluent and precise, the structure of the paper could be improved for clarity. Technical terminology and mathematical expressions are correctly used and defined. Certain sections of the text, figures, and tables would benefit from clarification or streamlining, as detailed below. The references are sufficient and appropriate, and the supplementary materials are well-prepared.

I particularly enjoyed reading the discussion and would like to congratulate the authors on this aspect of the manuscript. I also appreciate the efforts made to account for uncertainties in both the proxies and the age model. However, I found the methods and results sections somewhat difficult to follow. As outlined below, my main concerns relate to the following aspects:

- **Age tie-point assignment:** How were these determined? Were peaks and troughs used? Inflection points would be preferable, as they are less sensitive to noise and aliasing.
- **Age references:** Different age references appear to have been used for visual alignment and age model revision. Are there systematic differences between records based on the LR04 stack versus the Greenland temperature record?
- **Definition of pre-industrial (PI):** The definition is unclear, particularly for seasonally biased proxies. If annual means are used for seasonal proxies, the resulting anomalies could be misleading.
- **Results presentation and description:** In several instances, I interpreted the figures differently than described. To aid clarity, it would be helpful to outline at the beginning of the Results section the specific aspects being examined, and focus the result description on those. The manuscript is already quite long, and much of the detailed description of individual / regional patterns is not discussed later. I suggest moving the current sections 3.1 to the Supplementary Information and replacing them with a more focused summary of the main results that are subsequently discussed in section 4.

I hope these comments are constructive and helpful in revising the manuscript.

We thank the referee for this brief description and constructive comments.

In the revised version of the manuscript, we have improved the description of the methods used in this study, move the section 3.1. to the supplementary material, clarify some relevant points (e.g. PI definition). We have replaced the current section 3.1 (i.e. description of each individual records) by a comparison between different stacks, using all records, excluding $\delta^{18}O_p$-based reconstructions, or keeping only selected records. We have also re-calibrated all SST derived from $\delta^{18}O_p$ by using the Breitkreuz et al. (2018) database for $\delta^{18}O_{sw}$. We have revised our stacking

method using a smaller maximum range for latitudinal mean (max=10° now, the previous was 20°) and removing the additional +/- 0.5°C for lab uncertainty (not suggested by the referee but helpful to stabilize the stacking process).

**Age-Tie point assignment**: this point is discussed below. Briefly, we used peaks and troughs as they visually produced the "best fit" with the reference record. Mid-slope or inflection points tended to slightly shift a few events (e.g. interglacial peak, glacial maximum).

**Age reference**: We did not test to align our records to the LR04 reference. In terms of median age, we have no doubt that the two methods can produce similar alignment. However, the age error of the LR04 reference is ± 4 ka, while the mean absolute error of AICC2023 is ~ ± 1 ka. Adding this absolute error in the Bayesian age model process would result in a very large final uncertainty and, therefore, include larger uncertainties in the final estimate.

**Definition of the PI**: This point is discussed in the responses below. Seasonal SST anomaly are calculated with the corresponding monthly PI SST (as HadISST database gives SST for each month).

**Results presentation and description**: We have moved the section 3.1 to the SI in order to be more focus on global and regional patterns.

Find below our point by point responses.

**Specific comments**

**Line 12**: "…the fact that is associated…" — It appears that "it" is missing; please revise for clarity.

We have added a "it" in this sentence. —Line 12.

*"[...] from the fact that it is associated with the highest atmospheric $CO_2$ concentrations [...]"*

**Line 14–15**: The sentence "…it is challenging to assess the regional and global patterns of climate variability…" is quite long and requires several readings to grasp the context. Consider rephrasing for clarity and conciseness.

We rephrase this sentence to improve the clarity. —Lines 14-16.

*"[...] interval, yet synthesizing them into consistent regional- and global-scale climate signals is challenging because they are scattered across the globe and based on heterogeneous chronological frameworks."*

**Line 78**: "magnesium and calcium" should be revised to "magnesium-to-calcium."

We have modified this sentence. Note that this paragraph is completely revised, giving more details about the SST proxies. —Line 84.

*"The magnesium-to-calcium (Mg/Ca) ratio of [...]"*

**Line 79**: "Tex" should be capitalized to "TEX." Please ensure consistent use throughout the manuscript.

We have modified Tex to TEX throughout the manuscript. Note that this paragraph is completely revised, giving more details about the SST proxies.

**Line 79–83**: "old SST data" — Suggest rephrasing to "existing SST data" to better reflect that the key issue is differences in calibration methods rather than data age.

We have modified the term "old" to "existing". Note that this paragraph is completely revised, giving more details about the SST proxies. —Line 99.

*"[...] highlight the need to recompute existing SST records with harmonized [...]"*

**Line 95–98**: It is unclear whether the 98 records cited are those included after or before applying the selection criteria. Please clarify this point.

We have clarified this point in the sentence. —Lines 113-114.

*"[...] from marine sediment cores, corresponding to those retained after applying our selection criteria, to reconstruct both [...]"*

**Line 98–99**: "propose to include SST" — Suggest revising to "decided to include SST," since this reflects the authors' actual methodological decision.

We have modified "propose" to "decided". —Line 117.

*"We decided to include SST [...]"*

**Line 103–105**: The sentence is somewhat confusing, implying TEX86 reconstructs SST but is associated with subsurface temperatures. Consider rephrasing.

We have modified the sentence to improve clarity. This paragraph is now in the introduction. —Lines 93-94.

*"It can estimate SST through a temperature-dependent change in ring structures, but often reflects subsurface conditions (e.g. Schouten et al 2002; Lopes Dos Santos et al., 2010; Tierney & Tingley, 2014; Rouyer-Denimal et al., 2023), limiting its comparison with surface-based SST reconstructions."*

Additionally, since it is well established that TEX86 can reflect subsurface conditions, please consider citing earlier studies that first demonstrated this, rather than more recent ones.

We have added earlier studies as references for this statement. This paragraph is now in the introduction. —Line 94.

*"It can estimate SST through a temperature-dependent change in ring structures, but often reflects subsurface conditions (e.g. Schouten et al 2002; Lopes Dos Santos et al., 2010; Tierney & Tingley, 2014; Rouyer-Denimal et al., 2023), limiting its comparison with surface-based SST reconstructions."*

**Line 112–113**: The connection between the cited studies and the "efficiency" of Bayesian calibrations is unclear. Consider rephrasing to state that these studies *applied* the Bayesian calibrations, rather than demonstrated their efficiency.

We have modified this sentence to improve clarity. —Line 129-130.

*"This approach was previously applied in SST syntheses [...]"*

**Line 116–117**: Seasonal production is not generally cited as a reason for UK′₃₇ saturation. Please revise or clarify this claim.

We have removed this statement. —Line 135.

*"Slope attenuation is an important feature in high-temperature [...]"*

**Line 121–124**: The Mg/Ca–temperature relationship is not linear. Please revise accordingly.

We have modified this sentence. —Line 139.

*"[...] or seawater pH, the original calibration between Mg/Ca value and temperature (e.g., Anand et al., 2003) is outdated."*

**Line 133**: "dimensions of about N × 1000" — Some clarification is needed on the variability of this number. Why not use N × 1000 as for other proxies? Is it a small variation (e.g., 999–1001) or more substantial (e.g., 900–1100)?

The calibration code initially produces a N x 2000 matrix. We resample this matrix to fit with the typical N x 1000 format of the other calibrations to facilitate the data analyzes (anomaly calculations, stacking process). We have added a short sentence at the end of this paragraph. —Line 149.

*"[...] with dimensions of about N×1000 to be in line with the other proxies."*

**Line 136**: Consider using *Breitkreuz et al. (2018, JGR Oceans)* instead of *LeGrande and Schmidt (2006)*, as it is more recent and may be more appropriate here.

We have modified our code and we now use the Breikreuz et al. (2018) database. We have removed the LeGrande and Schmidt (2006) citation and reference. —Lines 152-153.

*"[...] we used the modern $\delta^{18}O_{sw}$ from Breitkreuz et al. (2018) to run these calibrations."*

**Line 137–139**: More detail is needed about the ice volume correction — specifically, how large is it?

We applied the ice-volume correction as suggested by Malevich et al. (2019) with the equation described in the Appendix B of their paper:

$$\delta_{corrected} = \frac{1000 + \delta_{calcite}}{1 + \frac{\delta_{ice}}{1000}} - 1000$$

As we use LR04 as reference for $\delta^{18}O_{ice}$, the correction is between ~3 and ~5 per mil. We have added a reference to the original study—Line 153.

*"A simple ice volume correction following Malevich et al. (2019) was applied before calibration, using the benthic δ¹⁸O stack LR04 (Lisiecki and Raymo, 2005) as a reference for global ice-volume changes."*

**Line 139**: "other published SST records" — Does this refer to geographically nearby records? If so, please define "nearby" more precisely.

We have added "from the same core" to improve clarity. —Line 155.

*"[...] using other published SST records from the same core (published mean) if available, or [...]"*

**Section 2.3**: This section is somewhat difficult to follow and warrants improvement, especially given its importance for anomaly calculations.

- How were the 1000 proxy values derived? Were HadISST values used as inputs to the Bayesian calibrations, and if so, how were these converted back to SSTs?

The initial SST (from HadISST) was converted to a 1000 proxy values by applying a forward Bayesian model. Then, the median proxy value was calibrated to obtained a 1000 SST values corresponding to the modeled estimate of the proxy value. We have added a few sentences to improve clarity.

*"[...] by forward modelling (with the Bayesian calibrations previously described) the SST from [...]"*— Lines 173.

*"For MAT reconstructions, an ensemble of 1×1000 probable PI SST values is generated from a normal distribution using the mean SST and standard deviation derived from the HadISST database (Rayner et al., 2003)."* — Lines 178-180.

- What pre-industrial (PI) references are used for seasonal proxies? If annual mean SST is used for proxies with seasonal production, this may introduce bias to the anomaly calculation.

The HadISST database gives monthly SST. We take the mean of the corresponding SST (summer, winter or other for specific cases) for our seasonal SST. We have added a short sentence to clarify. —Lines 178-179.

*"The same process was applied to seasonal proxies, taking only the corresponding monthly mean SST from the HadISST database."*

- Since many Pleistocene records lack modern/PI sediments, an alternative would be to use published core-top proxy values from the same grid cell (e.g., those from global calibration databases).

We are aware that core top global database have advantages, but also drawbacks as it can represents present-day SST. As we use the PI period from 1870 to 1899 to have the more appropriate SST anomaly to compare with climate model simulations, we think that the use of the HadISST is the best way to reduce biases in the process. Therefore, we decide to not use this

alternative. We have added two sentences in section 4.1 to discuss about this limitation. — Lines 417-420.

*"The use of core-top database could also be considered, since proxy values are directly available. However, these records mainly reflect a "present-day" conditions, which may differ from a PI reference and thereby introduce biases in final global- and regional-scale temperatures estimates."*

**Line 160**: "limit" → "limitation"

We have modified this word. —Line 179.

*"However, we are aware of the limitation of this PI [...]"*

**Line 160**: The phrase "discrepancies of the HadISST database" is unclear. Consider clarifying what discrepancies exist within the dataset or relative to.

We have clarified this sentence. —Lines 179-180.

*"[...] discrepancies between model ensembles and the HadISST database can exist, especially [...]"*

**Line 176**: "propose to use of" — Please rephrase for grammatical correctness.

We have corrected this sentence. —Line 195.

*"[...] we propose to use the Greenland synthetic [...]"*

**Line 178**: "to account the 'bipolar seesaw'" — Consider rephrasing for clarity, e.g., "to account for the effects of the bipolar seesaw."

We have modified this sentence. —Line 197.

*"[...] in order to account for the effects of the "bipolar seesaw" observed [...]"*

**Line 193**: Typo — "planctik" should be corrected.

We have corrected this word. —Line 213.

*"[...] (i.e. $\delta^{18}O$ of planktic foraminifera after removing [...]"*

**Line 196–198**: This is a critical methodological step and needs more detail:

Please clarify the criteria used to select age tie-points. Based on the supplementary figures, it appears that peaks and troughs were used. However, it is more common in the community to use inflection points, as they are less susceptible to aliasing and more robust against noise, particularly in lower-resolution records.

At the beginning of this project, we have compared three alignment method: peaks and troughs (uses in this study), inflection points and mid-slope. The visually "best fit" with the reference records has been attained with the peaks and troughs method. Moreover, as we visually define an "alignment error", it is easier to determine it with peaks and troughs than with inflection points or mid-slope.

Therefore, we decide to keep our alignments with peaks and troughs and clarify the sentence. —Lines 216-217.

*"Once "tie-points" were defined using peaks and troughs (Fig. S2), we visually estimated [...]"*

The authors wrote that uncertainty estimates were visually assigned but it is not immediately clear to the reader how. I suggest indicating these uncertainties in the supplementary figures and including them in a table in the SI.

All tie points and prior uncertainties will be made available in a Zenodo repository upon publication of the manuscript. We have added a sentence about the age uncertainty and how we visually define them. —Lines 219-220.

*"The age uncertainties are most of the time estimated as half of the duration of the event (peak or trough)."*

**Line 199**: Please clarify how "basin references" are defined / selected.

We have added a short sentence to clarify how these basin references were selected. —Lines 211-212.

*"Four "basin references" were defined, based on their high-resolution and the ability to align those records to reference records (i.e. ice-core records are synthetic curve)."*

**Line 204–205**: See comment above on Line 196–198.

We have clarified the method to visually define the uncertainty in lines 219-220 and refer to these lines in this sentence. —Line 226.

*"For each method, the age and depth uncertainties are visually defined as described above."*

**Line 233–234**: It's not necessarily a drawback that high-resolution records contribute more to the stack. One might argue that such weighting improves the robustness of the stack. Consider clarifying this point or adjusting the framing.

We have modified this sentence. —Lines 252-255.

*"However, their procedure has two major drawbacks: (1) the PI from the HadISST database (Rayner et al., 2003) does not reflect the SST from a proxy and have not any uncertainties as it is a single fixed value; (2) the non-uniform geographic distribution of their records induces a bias in the global mean temperature if a large number of grids are concentrated into a single latitudinal band."*

**Line 237**: "inspired from" should be revised to "inspired by."

We have modified this sentence. —Line 257.

*"[...] largely inspired by Osman et al. (2021) [...]"*

**Line 240–241**: The phrasing "random redrawn" and "different age-SST" is awkward. Please consider rephrasing for clarity.

We have modified this sentence to improve clarity. —Lines 260-261.

*"The first step of the Monte-Carlo process consists of randomly resampling the $N^{th}$ age-SST ensembles to generate new age-SST pairs in each iteration."*

**Line 243–245**: This description is difficult to follow. Does "random grid mean" refer to the averaging of values within a grid cell or the shifting of grid boundaries? The rationale for randomly adjusting the latitudinal mean is unclear. Also, the 20° range seems quite large and may blur distinctions between tropical and extratropical settings. Please clarify.

We have clarified this paragraph to better explain the process. We have decided to reduce the maximum range of latitudinal bands from 20° to 10° and the result is not very different compared to the previous one (the only difference is associated with the new $\delta^{18}O_p$-based SST using the $\delta^{18}O_{sw}$ database from Breikreuz et al., 2018). —Lines 261-264.

*"The second step, applied to each 500-year time bin, we first compute site means (averaging all data from ensembles available for this time-bin within a site, regardless of the proxy). These are then aggregated into grid-cell means, using randomly defined grid sizes (2 - 5°), and subsequently into weighted latitudinal means, using bands of randomly varying width (2.5 - 10°)."*

**Section 3.1**: This section is lengthy and contains descriptive detail that is not followed up in later sections. I recommend summarizing key observations and moving some site- and region-specific details to the Supplementary Information. Additionally, please clarify:

- How is glacial maximum defined? Lowest temperature within MIS 10?

Yes, the glacial maximum is defined here as the lowest temperature during MIS 10. We have added a short summary of the terms used in these description at the beginning of the section (now in the SI).

- How is the beginning of deglacial warming defined — change-point analysis, or visual inspection?

We have defined the deglacial warming as the "start" of warming defined after visual inspection (now in the SI).

- What is the distinction between "peak" and "plateau" in terms of their duration?

A peak is defined as one or two data point(s) (i.e. 500 to 1,000 years) showing the highest temperatures. A plateau is all periods longer that a peak. We have added detail about the terms used in these descriptions (now in the SI).

- For marginal seas, some description glosses over differences between proxies and records. Consider explicitly stating which proxies/records show which patterns.

We have modified the description of these marginal seas (now in the SI).

**Line 265–267**: These records are based on different proxies, which could explain the discrepancies.

We have clarified the sentence (now in the SI).

**Line 281**: MD97-2141 is located in the western Pacific, not in the South China Sea.

We have removed MD97-2141 of this sentence (now in the SI).

**Line 290**: The definition of "glacial maximum temperature" is unclear; this made it difficult to verify the interpretation based on the figure.

We have clarified this term at the beginning of the section 3.1. (now in the SI in the revised version of the manuscript).

**Line 333**: Consider marking the start date in the figure to assist the reader.

Each subplot of these figures is already very rich in information. We think that adding a new symbol (or more if there are different proxy-based SST) would be counterproductive and make the figure difficult to read.

**Line 342–343**: This sentence is unclear; please revise.

We have modified this sentence (now in the SI).

**Line 348**: Site MD02-2575 does not show large glacial–interglacial oscillations.

We have added details in this sentence to improve clarity (now in the SI).

**Line 351**: The $\delta^{18}O_p$ record does not appear to show an abrupt deglacial warming. Please revise accordingly.

In this sentence, the brackets "...($U^{K'}_{37}$-based SST record)..." indicate that we are referring to this record, not to the $\delta^{18}O_p$-based SST (now in the SI).

**Line 358–360**: Same as above — the patterns described are not apparent in the $\delta^{18}O_p$ record.

We have added that we refer to the $U^{K'}_{37}$ record (now in the SI).

**Line 361–365**: This highlights the need for a clearly stated method/criteria for visual assessment. For PRGL-1, $\delta^{18}O_p$ shows decreasing temperatures during deglaciation, in contrast with the $UK'_{37}$ record. If these are considered consistent, why are similar discrepancies in the Caribbean interpreted as strong disagreements?

In this paragraph, we highlight a strong warning that the referee may have probably misses: "...with the exception of the "cold" periods (i.e. Glacial maximum and the end [...] where the $\delta^{18}O_p$-based SST record exhibits anomalously warmer temperatures".

Except this period, the two SST records are consistent in term of variability, with minor amplitude of change in the $\delta^{18}O_p$-based SST record. It is different with Caribbean sites, where the two records are completely different in terms of amplitude and variability for the two sites.

No modifications here.

**Line 372**: The MAT record at ODP-1089 is incomplete and does not clearly show deglacial warming.

We agree with the referee that missing data in the MAT record is an issues because it does not cover the interglacial, however, $\delta^{18}O_p$-based SST does.

No modifications here.

**Line 373 & 395**: While I agree that ODP-1084 shows a plateau, to my eyes MD96-2094 appears to show a peak, more similar to MD97-2120.

Please consider defining "plateau" and "peak" at the beginning of the section.

We have added a definition of "peak" and "plateau" at the beginning of the section 3.1 (now in the SI).

**Line 449**: Do the stated uncertainties correspond to the shaded envelope in Figure 5c? They appear slightly different.

Uncertainties are defined as half of the difference between the 99th and 1st percentiles. As the stacking process has been modified. We have adjusted all values and uncertainties (defined now as $1\sigma$) throughout the manuscript (now Fig. 4 in the revised manuscript).

**Line 450–455**: It was difficult to identify the patterns described in Figure 5. Consider marking key transitions in the figure.

We have identified the periods described in the description in the panel B (now Fig. 4).

**Line 453**: Is the resolution sufficient to detect such a small change over 2 kyr intervals?

The average temporal resolution is ~1,700 years (line 117) and the methodological approach (ensemble of age-SST pairs) allows a full cover of the MIS 9-10 interval. The median estimate is based on a process taking into account all source of uncertainties. Therefore, we find that this is sufficient to estimate such a small change over 2 kyr.

**Line 487–489**: This depends on how "deglacial warming" is defined. If it's defined as a significant shift, there would not be multiple discrete steps.

We have modified this part of the section as we define the onset of the deglacial warming as a significant shift. Lines-356-357.

*"The deglacial in the North Pacific occurs at ~338.5 ka after a ~10 ka-long phase of stability, with a warming of ~3.4 °C over 7 ka."*

**Line 500**: It would be helpful to include in the SI a figure showing all records used to construct the basin/hemisphere stacks.

All records used to construct the basin / hemisphere stacks are already shown in Figs 3-4 (now in the supplementary material as Fig. S1-S2) as "annual temperatures". As we now use stacks with "selected" records, we have added a short paragraph at the end of the SI S1 (S1.8) to indicate the "excluded" records.

**Line 509–510**: Please indicate where the $\sigma$ uncertainty values can be found and how they were calculated.

All stacks data (i.e. percentile from 1 to 99) are available in a Zenodo repository. The standard deviation is derived from these datasets (half of the difference between the 84th and 16th percentiles). We have added a short sentence at the beginning of the section 3.2. —Lines 305-306.

*"Errors in Celsius degree are given as the $\sigma$ error."*

**Line 533–535**: Have the authors considered using the *Breitkreuz et al. (2018)* $\delta^{18}O_{sw}$ dataset? The described unexpected temporal evolution raises the question of whether including $\delta^{18}O_p$ records truly adds value to the compilation beyond increasing record count.

We have modified our dataset for $\delta^{18}O_{sw}$. We originally used LeGrande and Schmidt (2006) as recommended in the BAYFOX documentation. As a result, high-latitudes $\delta^{18}O_p$-based SST reconstructions appear colder, while low-latitudes records appear warmer.

As a test, the authors could create stacks with and without $\delta^{18}Op$ to assess their impact on the results.

We have added a sensitivity tests by stacking data without $\delta^{18}O_p$ (now section 3.1). The number of records drastically decrease (from 77 to 41) and the estimated stacks exhibit warmer temperatures (at global, regional and hemispheric scales). This difference may occur for two reasons: 1) $\delta^{18}O_p$-based SST records systematically pull down the stack values, indicating a bias in calibration (e.g. $\delta^{18}O_{sw}$ changes could be important depending on the area, ice-volume correction under/over estimated); 2) Reducing almost half of the number of records means that there is not enough information to produce a sufficiently accurate global or regional pattern. This part is now discussed in section 3.1. —Lines 273-294.

*"3.1 Global and regional sensitivity tests*

*Based on the individual description of each records (Supplementary Information S1), we identified some discrepancies between SST reconstructions at a same site (i.e. with different proxies) or within the general tendency of a region. Most of these inconsistencies are linked to $\delta^{18}O_p$- or, to a lesser extent, Mg/Ca-based SST, suggesting a lack of environmental context prior to the calibration (see section 4.1.1 for further discussions). To evaluate the impact of such records on the stacking procedure, we conducted three sensitivity tests: in the test (1), we include all annual SST records (n=77); in the test (2), we exclude all $\delta^{18}O_p$-based SST reconstructions (n=41), which often display lower temperature values or distinct pattern of variability; in the test (3), we retain only records showing a consistent range and pattern of variability of records located in the same area, leading to the exclusion of 13 records (n=64) primarily based on $\delta^{18}O_p$ and Mg/Ca proxies. The selection in this third test is based on a visual assessment.*

*Overall, the resulting stacks display very similar variability across tests, regardless of the method applied. At global and hemispheric scales, the amplitude and timing of major climatic features (e.g. Deglaciation, interglacial optimum, glacial inceptions) remain unchanged (Fig. 3). At basin scale, however, the South Atlantic and Indian stacks without $\delta^{18}O_p$-based SST records (second test, Fig. 3H, J) show differences in the shape of variability. The main differences between tests are related to the absolute temperature values and amplitude: stacks excluding $\delta^{18}O_p$ records are systematically warmer (~1 to 2 °C), with larger hemispheric amplitudes, and the North Atlantic shows particularly cold MIS 10 conditions in this case.*

*The test including all records (Fig. 3A-E) is broadly consistent with the others but results in lower SST values, indicating that inconsistent records tend to pull down the stacks and potentially bias the reconstructions. Excluding $\delta^{18}O_p$-based SST (second test, Fig. 3F-J) produces the largest shift in values, but also strongly reduces the number of records. The "selected records" test (Fig. 3K-O) appears to provide a reasonable compromise: it excludes outlier estimates without systematically discarding one proxy type, while retaining a sufficient number of records to build robust stacks. We therefore adopt this third stacking approach for the analyses and interpretations that follow."*

**Line 535–538**: Consider sensitivity tests using the *Gray and Evans (2018)* multivariate calibration to explore the influence of non-thermal factors.

Before conducting our calibrations, we consulted with W. Gray, who advised that either his calibration or the Bayesian framework of Tierney et al., (2019) could be applied. Since both approaches rely on similar prior information and assumptions, we chose to adopt the Tierney et al. (2019) calibration for consistency with the other proxy calibration approaches. A comparison of both calibrations would be interesting, but it is out of the scope of this study.

As material for future studies, all Mg/Ca original data will be published in a Zenodo repository if anyone wants to compare the two approaches.

**Line 543–544**: Please show the effect of including these records in the SI. Including problematic records may weaken the integrity of the compilation — this tradeoff merits discussion.

We ran a sensitivity test excluding 13 "problematic" records (from 77 to 64 records used in the stacking process). We have added the results and a brief discussion in the new section 3.1. The variability is unchanged without these records and only the stack values are changed, generally towards warmer temperatures. —Lines 273-294. See above (comment "**Line 533–535**" for the new section 3.1.

**Line 545–546**: The PI reference remains unclear. An alternative would be to compare with climatological products like the *World Ocean Atlas*.

WOA only proposes present-day SST, that is different of the PI SST, much more suitable for future data-model comparisons. We have decided to not perform this alternative to keep a consistent synthesis in line with future studies. We have added two sentences in the section 4.1.1. —Lines 417-420.

*"The use of core-top database could also be considered, since proxy values are directly available. However, these records mainly reflect a "present-day" conditions, which may differ from a PI reference and thereby introduce biases in final global- and regional-scale temperatures estimates."*

**Line 568–569**: This sentence is confusing. Do the authors mean basin-scale temperature trends should be considered independently from global averages? Please clarify.

We have clarified this sentence. We mean that to have a comprehensive interpretation of T-IV and MIS 9 changes, we need to take all reconstructions and estimates (from individual SST records to global stacks). —Lines 436-438.

*"Accordingly, a comprehensive interpretation of temperature changes across T-IV and MIS 9 requires integrating information from individual records, as well as from regional, hemispheric, and global SST stacks."*

**Line 582–583**: Please elaborate — what is the benchmark, and what makes this anomaly stand out?

There is no benchmark needed here, we just want to highlight the fact that this mechanism appears clearly during T-IV and is the main driver of the early warming in the South Atlantic. We have modified this sentence to improve clarity. —Lines 451-452.

*"This mechanism is particularly evident during T-IV, where it amplifies SST variability and appears to be the main driver of the early warming observed in the South Atlantic basin."*

**Line 585**: The reasoning is unclear. Is the same mechanism present in other basins? Is the SH stack dominated by Atlantic records?

We have added a few words to improve the clarity of this sentence. —Lines 453-454.

*"As most of Southern Hemisphere records are located in the South Atlantic (see Supplementary Information S1), this may also explain [...]"*

**Line 602**: Please specify the latitude and month used for "solstice insolation."

We have added "65°N" in this sentence and in the Fig.6 (now Fig. 5 in the revised manuscript). —Line 472.

*"This early warming in the NH may result as a response to the 65°N summer solstice insolation [...]"*

**Line 615**: Should this refer to Figure 6D? Please check.

We have modified the reference to the Fig. 6D (now Fig. 5).

*"[...] temperature anomalies to Northern ones (Fig. 5D)."*

**Line 639**: Consider rephrasing "redistributing surface temperature" to "redistributing heat."

We have modified this part of the sentence. —Line 513.

*"This seesaw pattern underscores AMOC's critical role in redistributing heat (e.g. Stocker and Johnsen, 2003; J. Lynch-Stieglitz, 2017; Pedro et al., 2018; Davtian and Bard, 2023)."*

**Line 677**: Typo — "global temperature similar" should be revised.

We have revised this sentence. —Line 550.

*"[...] the previous stacks exhibit global temperature similar between them (from 0 to 0.3 relative to PI), while our new [...]"*

**Line 682–683**: Are Holocene anomalies directly comparable to MIS 9e results? Were all anomalies calculated using the same reference period?

In this sentence, we refer to the Shakun et al. (2015) MOT estimates, that are based on anomalies relative to the mean Holocene. This is not really comparable with our results, but gives a range of values to initiate the discussion about the differences between GSST and MOT.

No modifications here.

**Line 741**: Why? Is it due to differences in the spatial coverage of these compilations? Please clarify.

We have modified this sentence to improve clarity. —Lines 616-617.

*"This discrepancy may reflect differences in temporal phasing between interglacial peaks in different regions."*

**Line 769–771**: This sentence is unclear. Does the "last 60 years" refer to the *Osman* stack or *HadCRUT5*? Please clarify.

We have added the reference "Morice et al., 2021" in this sentence to improve clarity. Line 645.

*"Notably, for T-I (Osman et al., 2021), which has the highest-resolution GMST stack, the temperature increase over the last 60 years (Morice et al., 2021) is almost twice as high [...]"*

**Figures**

**Figures 3 & 4**: Please indicate whether sites are located in the east, south, or marginal zones of each basin.

We don't understand the referee's comment: the location of each site is already indicated in each subplot. We have increased the font-size to improve clarity (now in the SI).

**Figure 5**: Please mark MIS 10.

We have added "MIS 10" in the figure (now Fig. 4 in the revised manuscript).

**Figure 6D**: Typo in y-axis label — "transfert" should be "transfer."

We have modified this mistake (now Fig. 5 in the revised manuscript).

---

## Author Response (AR2)

Saint-Martin d'Hères, September 22th 2025.

**Dear Editor B. Risebrobakken,**

We thank you for your constructive comments on the revised manuscript. We have improved clarity and consistency regarding the three points raised and corrected minor typos. The modifications are as follows:

- Lines 57 & 76: typo corrections.
- Lines 218-221: added three sentences to explain the choice of the 'peak and trough' method.
- Lines 424-425: added a short sentence about the limitations of core-tops records.
- Line 537. We clarified the sentence.
- Line 729. We added the link to the Zenodo repository, where all individual SST and tie-points data are available, along with the stack results (including the three results shown in Fig. 3).

Please find the revised manuscript attached with track changes (in red).

Best regards,

**Stevenard Nathan**

Corresponding author Institut des Géosciences de l'Environnement (IGE) 54 Rue Molière. St-Martin d'Hères - France.

E-mail: nathan.stevenard@univ-grenoble-alpes.fr